# Source-explicit estimation of brown carbon in the polluted atmosphere over North China Plain: implications for distribution, absorption and direct radiative effect

Jiamao Zhou[1,2], Jiarui Wu[1], Xiaoli Su[1], Ruonan Wang[1], Imad EI Haddad[4], Xia Li[1], Qian Jiang[1], Ting Zhang[1], Wenting Dai[1], Junji Cao[3], Andre S.H. Prevot[4], Xuexi Tie[1], Guohui Li[1]

[1]State Key Laboratory of Loess Science, Institute of Earth Environment, Chinese Academy of Sciences, Xi'an 710061, China

[2]National Observation and Research Station of Regional Ecological Environment Change and Comprehensive Management in the GuanZhong Plain, Xi'an 710061, China

[3]Institute of Atmospheric Physics, Chinese Academy of Sciences, Beijing 100029, China

[4]PSI Center for Energy and Environmental Sciences, Paul Scherrer Institute, 5232 Villigen, Switzerland

*Correspondence to*: Guohui Li (ligh@ieecas.cn)

**Abstract.** Brown carbon (BrC) plays a significant role in altering atmospheric radiation. Beyond biomass and biofuel combustion, recent studies identify fossil fuel sources—especially residential coal burning and vehicle exhaust—as major contributors to BrC. This underscores a gap in climate models, which often assume fossil fuel organic aerosols (OA) are non-absorbing or treat all OA as light-scattering. In this study, we simulate BrC over the North China Plain (NCP) during a winter pollution event using the WRF-Chem model, incorporating explicit BrC absorption properties. The model aligns well with observed pollutant and aerosol levels, revealing an average near-surface BrC concentration of 5.2 $\mu$g m$^{-3}$, contributing 16.4% to aerosol absorption at 365 nm. Using a diagnostic adjoint approach, we estimate that BrC exerts a direct radiative effect (DRE) averaging -0.09 W m$^{-2}$ at the top of the atmosphere, reducing the cooling effect of organic carbon by 28.0% and producing a local warming effect of up to +0.40W m$^{-2}$. Coal combustion is the largest BrC source in the NCP in 2014, though secondary BrC also significantly impacts the regional radiation balance.

**Key words:** Brown Carbon, emission sources, absorption, direct radiative effect

## 1 Introduction

Brown Carbon (BrC) is a collective component for those colored organic compounds with wavelength dependent light-absorption properties (Mukai and Ambe, 1986; Kirchstetter et al., 2004; Andreae and Gelencsér, 2006). BrC has been recognized as an important short-lived climate forcer contributing considerably to climate change by warming of the atmosphere (IPCC, 2013; Feng et al., 2013; Jacobson, 2014; Jo et al., 2016; Brown et al., 2018). A study suggested that the light absorption induced by BrC can be equal to or even higher than that of black carbon (Pokhrel et al., 2017), and substantially influences atmospheric radiative forcing. Recent studies have shown that BrC accounts for 30%~50% of the total absorption of aerosols in Atlanta USA, Brazil and Hebei China (Hoffer et al., 2006; Yang et al., 2009; Liu et al., 2013). The direct radiation effect (DRE) caused by BrC is greater than +1 W m$^{-2}$ in some regions, such as South Asia, Africa and Southeast Asia, much higher than the global average (Park et al., 2010; Feng et al., 2013; Lin et al., 2014; Saleh et al., 2014; Jo et al., 2016; Wang et al., 2018; Yan et al., 2018). However, the modelled DRE associated with BrC remains highly uncertain, with variations spanning an order of magnitude difference. In particular, the estimated global DRE of BrC is in the range between +0.03 W/m$^2$ to +0.57 W/m$^2$ (Hammer et al., 2016), which is caused by the limited observations of BrC mass and absorption properties (Tuccella et al., 2020; Saleh, 2020).

It has been well established that BrC is not a single substance, but a general term for light-absorbing organic aerosols. A series of laboratory measurements and observations in the earlier years demonstrate that BrC is mainly associated with smoldering biomass burning (BB) or biofuel (BFs) combustion (Chakrabarty et al., 2010; Chen and Bond, 2010; Lack et al., 2012; Washenfelder et al., 2015; Kumar et al., 2018). On the other hand, OA from fossil-fuel combustion are generally assumed to be non-absorbing as the combustion conditions for fossil fuels (FFs) are typically not conducive for BrC formation (Hecobian et al., 2010; Shapiro et al., 2009; Bond et al., 2013). Therefore, earlier climate model studies have assumed that primary OA from BB and BFs combustion is the main or sole BrC source (Feng et al., 2013; Jacobson, 2014; Saleh et al., 2014; Hammer et al., 2016; Brown et al., 2018). Recent studies have also incorporated the ageing of secondary organic aerosol (SOA) (Jo et al., 2016; Wang et al., 2018; Zhang et al., 2020). However, more recent exceptions are being found in low-efficiency residential-coal combustion (RCC) (Bond, 2001; Yan et al., 2017; Xie et al., 2019; Tian et al., 2019; Zhang et al., 2022a) and fuel-oil combustion in vehicle and ship engines (Xie et al., 2017; Corbin et al., 2019; Tang et al., 2020; Huang et al., 2022). It is now generally accepted that the formation of BrC is not exclusively linked to the chemical make-up of biomass fuels but is most critically determined by the combustion conditions (Saleh et al., 2018; Cheng et al.,

2020; Saleh, 2020; Wang et al., 2022a). The key factor contributing to the high levels of BrC observed from biomass fuels is their combustion under relatively low-temperature and fuel-rich conditions, which are highly favorable for BrC formation. In contrast, fossil fuels, such as those burned in internal combustion engines, typically undergo combustion at higher temperatures and under more fuel-lean conditions, which are less conducive to BrC production (Saleh, 2020). China, as a developing country, coal is commonly used for residential heating in cold season, causing massive emissions of organic particles (Yan et al., 2017; Li et al., 2018). According to the National Bureau of Statistics of China (https://data.stats.gov.cn), the coal consumption in 2014 was about 4000 Tg, accounting for 65.8% of the total primary energy use of China. Of this, around 93 Tg is used as household fuel. The poor burning conditions and limited emission control facilities in this region could lead to substantial emissions of BrC. This could explain why, to date, all reported instances of coal-derived BrC have originated from China. Both Yan et a., (2017) and Mo et al., (2021) have used dual carbon isotope-based source apportionment method reported that fossil fuel, especially coal combustion from the residential sector is important source in northern China, even the largest contributor in some regions.

These recent findings indicate a critical gap on the treatment of BrC in chemical transport models, atmospheric chemistry models and climate models as well, which present an even greater concern, as they typically do not consider BrC at all (Ma et al., 2021; Jo et al., 2023; Gao et al., 2025; Ge et al., 2025). Addressing this gap requires expanding the scope of BrC sources, assigning distinct optical properties for each source and incorporating those that have been underrepresented or overlooked in past assessments into numerical models. In this study, we include the main primary emission sources (RCC, BB, BFs, FFs-TRA) of BrC and secondary derived BrC in a regional model, the Weather Research and Forecasting model coupled with Chemistry (WRF-Chem). A representative region, North China Plain (NCP), is chosen as the study domain with high anthropogenic carbonaceous aerosols due to the widespread use of coal and biomass burning for heating during winter and the increasing number of motor vehicles. We performed a month simulation to evaluate the surface distribution, absorption and the DRE of BrC in the NCP, by updating BrC optical properties of different sources. Sensitivity experiments have also been devised to assess the contribution of BrC from major sources.

**2 Model and Method**

**2.1 WRF-Chem Model and configurations**

The WRF-Chem model (Grell et al., 2005; Fast et al., 2006) modified by Li et al., (2010; 2011a; 2011b; 2012) is used to quantitatively estimate the BrC in the NCP. A heavily polluted month from January 1 to 30, 2014 is

selected for the simulation period. The anthropogenic emissions are developed by Zhang et al. (2009) and Li et al.
(2017), including five sources, namely agriculture, industry, power generation, residential, and transportation. The
biogenic emissions are calculated online using the MEGAN (Model of Emissions of Gases and Aerosol from
Nature) model developed by Guenther et al. (2006). Additionally, the grid-based RCC, BB and BFs combustion
emissions are used to update the BrC sources in this study, described later. The model simulation domain is shown
in Fig. 1. The detailed model description and configurations can be found in supplementary text S1 and Table S1,
respectively.

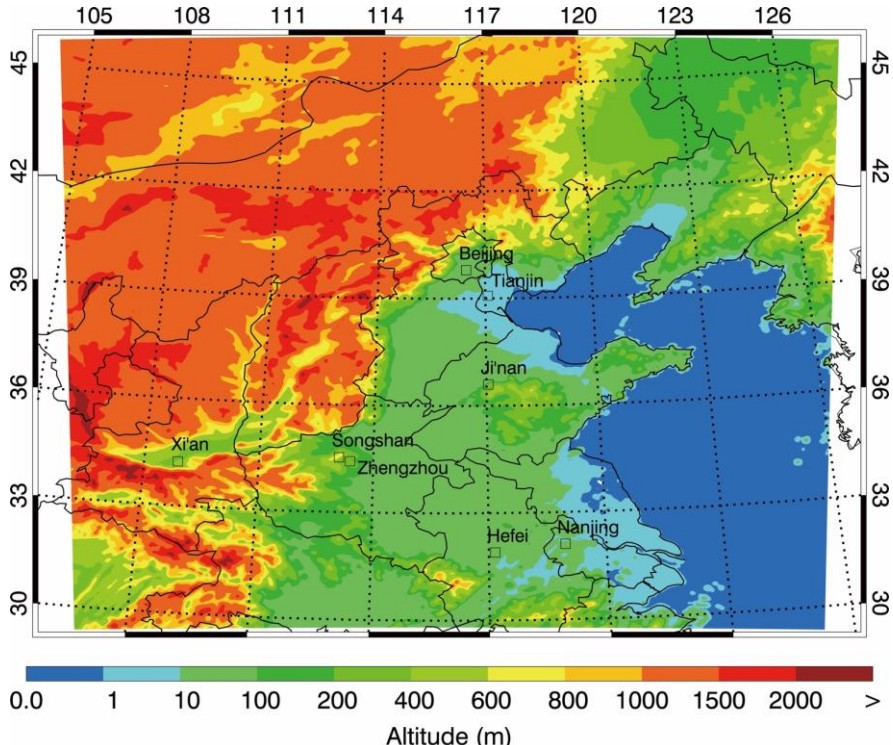


**Figure 1. WRF-Chem simulation domain with topography. The square denotes the field sites for simulation and**
**observation comparison**
**2.2    Aerosol radiative module**
The aerosol radiative module developed by Li et al., (2011a) has been incorporated into the WRF-Chem
model to calculate the aerosol optical depth (AOD or $\tau_a$), single scattering albedo (SSA or $\omega_a$), and the
asymmetry factor ($g_a$). In the aerosol module, aerosols are represented by a three-moment approach with a
lognormal size distribution:
$n(lnD) = \frac{N}{\sqrt{2\pi}ln\sigma_g}exp[-\frac{1}{2}(\frac{lnD-lnD_g}{ln\sigma_g})^2]$  (1)
Where D is the particle diameter, N is the number distribution of all particles in the distribution, $D_g$ is the
geometric mean diameter, and $\sigma_g$ is the geometric standard deviation. To calculate the aerosol optical properties,
the aerosol spectrum is divided into 48 bins from 0.002 to 20.0 μm, with radius $r_i$. The aerosols are classified
into four types: (1) internally mixed sulfate, nitrate, ammonium, hydrophilic organics and black carbon (BC), and
water; (2) hydrophobic organics; (3) hydrophobic BC; and (4) other unidentified aerosols (generally dust-like
aerosols). These four kinds of aerosols are assumed to be mixed externally. For the internally mixed aerosols, the
complex refractive index at a specific wavelength (λ) is calculated based on the volume-weighted average of the
individual refractive index. Given the particle size and complex refractive index, the extinction efficiency ($Q_e$),
$\omega_a$ and $g_a$ are calculated using the Mie theory at a certain wavelength (λ). The look-up tables of $Q_e$, $\omega_a$ and
$g_a$ are established according to particle sizes and refractive indices to avoid multiple Mie scattering calculation.
The aerosol optical parameters are interpolated linearly from the look-up tables with the calculated refractive
index and particle size in the module. The $\tau_a$ at a certain $\lambda$ in a given atmospheric layer $k$ is determined by
the summation over all types of aerosols and all bins:
$$\tau_a(\lambda, k) = \sum_{i=1}^{48} \sum_j^4 Q_e(\lambda, r_i, j, k) \pi r_i^2 n(r_i, j, k) \Delta Z_k \tag{2}$$
where $n(r_i, j, k)$ is the number concentration of $j$-th kind of aerosols in the $i$-th bin. $\Delta Z_k$ is the depth of an
atmospheric layer. The weighted-mean values of $\omega_a$ and $g_a$ are then calculated by using D'Almeida et al.,

119    (1991):

$$\omega_a(\lambda, k) = \frac{\sum_{i=1}^{48} \sum_j^4 Q_e(\lambda, r_i, j, k) \pi r_i^2 n(r_i, j, k) \omega_a(\lambda, r_i, j, k) \Delta Z_k}{\sum_{i=1}^{48} \sum_j^4 Q_e(\lambda, r_i, j, k) \pi r_i^2 n(r_i, j, k) \Delta Z_k} \tag{3}$$
$$g_a(\lambda, k) = \frac{\sum_{i=1}^{48} \sum_j^4 Q_e(\lambda, r_i, j, k) \pi r_i^2 n(r_i, j, k) \omega_a(\lambda, r_i, j, k) g_a(\lambda, r_i, j, k) \Delta Z_k}{\sum_{i=1}^{48} \sum_j^4 Q_e(\lambda, r_i, j, k) \pi r_i^2 n(r_i, j, k) \omega_a(r_i, j, k) \Delta Z_k} \tag{4}$$
When the wavelength-dependent $\tau_a$, $\omega_a$, and $g_a$ are calculated, they can be used in the Goddard shortwave
module.

124        It is worth noting that the aerosol liquid water content in the study is predicted with the inorganic

aerosols using a computationally efficient thermodynamic equilibrium model, ISORROPIA (version 1.7,
(Nenes et al., 1998; Fountoukis and Nenes, 2007). In this study, ISORROPIA is mainly used to predict the
thermodynamic equilibrium between the ammonium-sulfate-nitrate-chloride-water aerosols and their gas-phase
precursors $H_2SO_4$-$HNO_3$-$NH_3$-$HCL$-water vapor, and water uptake of aerosols is calculated using the Zdanovskii-
Stokes-Robinson (ZSR) correlation (Stokes and Robinson, 1966):
$$W = \sum_i \frac{M_i}{m_{oi}(a_w)} \tag{5}$$
Where $W$ is the mass concentration of aerosol liquid water (kg m$^{-3}$ air), $M_i$ is the molar concentration of
species $i$ (mol m$^{-3}$ air), and $m_{oi}(a_w)$ is the molality of an aqueous binary solution of the $i$-th electrolyte
with the same $a_w$ (i.e. relative humidity) as in the multicomponent solution.
The BrC in the model has an effective density of 1.2 g cm$^{-3}$ for primary BrC (Turpin and Lim, 2001) and of
1.0 g cm$^{-3}$ for secondary BrC (Hurley et al., 2001). The imaginary refractive index of BrC used in this study is
discussed in 2.3.2.

**2.3 Model modifications**

**2.3.1 Source separation of BrC**

The definition of BrC in the model is dependent on its sources. Due to the lack of BrC emission inventories,
According to the characteristics of energy structure in China, assumptions and code modifications of the WRF-
Chem model have been made to consider the BrC from different sources. These involve three separated primary
BrC sources, including BB emissions, FFs emissions from RCC and on-road vehicles (FFs-TRA), and a part of
SOA which has light absorption property whereas other types of primary OA (POA) and SOA are assumed to be
purely scattering. In this study, BB source corresponds to open fire, household biomass burning and biofuel
consumption emissions.
For the primary emissions, previous BrC simulations have substituted it with a proportion of POA directly
(Feng et al., 2013; Lin et al., 2014; Wang et al., 2014; Tuccella et al., 2020; Xu et al., 2024) , derived it from the
relationship between the burning efficiency and the observed aerosol light absorption (Jo et al., 2016; Zhu et al.,
2021), or determined it through parameterization where BrC absorption is a function of the BC-to-OA emission
ratio (Zhang et al., 2020). In the present work, we calculated the primary BrC emissions based on the bottom-up
OA emission inventory combined with reported annual BrC emissions from various primary sources, as shown in
Table 1. Firstly, we collected the reported annual emissions of BrC from RCC, BB and FFs-TRA by using bottom-
up inventory method. It should be noted that given the proximity of the study period (January 2014) to 2013, we
use the emissions of BrC from RCC and BB in 2013 provided by Sun et al., (2017; 2021), which is 592Gg and
712Gg, respectively. The emissions of FFs-TRA derived BrC is 76Gg, which is calculated based on the value of
2017 (Wang et al., 2022a)and scaled by a factor of 0.70 to reflect the change of annual civilian-owned motor
vehicles. We assume that the spatial and seasonal variation of BrC is similar to OA. Then bottom-up emissions
inventory induced monthly BrC emissions in the NCP in January 2014 is the annual BrC emissions multiplied by
the ratio of OA emissions in NCP *vs* China, and the ratio of OA emissions in January 2014 *vs* the whole year,
resulting in a value of 65.5 Gg, 56.8 Gg and 4.4 Gg for RCC, BB and FFs-TRA, respectively. Finally, the
proportion of the three primary emissions of BrC used in the model is 36.3%, 100.8% and 15.8%, respectively.
Figure 2 shows the contributing regions and burdens of the three separated primary sources of BrC.
**Table 1 The data for primary BrC emissions calculation**

| Primary sources of BrC | RCC | BB | FFs-TRA |
|---|---|---|---|
| Annual BrC emissions (Gg) in China | 592.0[a] | 712.0[b] | 76.0[c] |
| Ratio of OA emissions in the NCP *vs* China[e] | 57.7% | 51.0% | 69.4% |
| Ratio of OA emissions in January 2014 *vs* the whole year[e] | 19.2% | 14.0% | 8.3% |
| Bottom-up emissions inventory induced monthly BrC emissions in the NCP in January | 65.5 | 56.8[d] | 4.4 |
| Emissions in the NCP in January 2014[e] | 180.2 | 56.4 | 27.9 |
| BrC emissions ratio for primary sources used in the model | 36.3% | 100.8% | 15.8% |

[a] The BrC emissions from China's RCC in 2013 was reported by Sun et al., (2017) based on experiments involving seven coals
were burned in four typical stoves as both chunk and briquette styles.
[b] The calculated BrC emissions from China's household biomass burning in 2013 reported by Sun et al., (2021) using 11 widely
used biomass types in China burned in a typical stove.
[C] The estimated BrC emissions from vehicle exhaust in 2017 was 109 Gg reported by Wang et al., (2022a). In this study, the
emissions of FFs-TRA derived BrC is 76.0 Gg with a yearly scale factor 0.70 which derived by the annual civilian-owned
motor vehicles between 2014 and 2017.
[d] The value of BrC emissions in NCP in January 2014 is additionally added with OA emitted from the open-biomass burning
(6 Gg) which is assumed to be entirely light-absorbing.
[e] These values were derived from the OA emission inventory described in Sec. 2.1

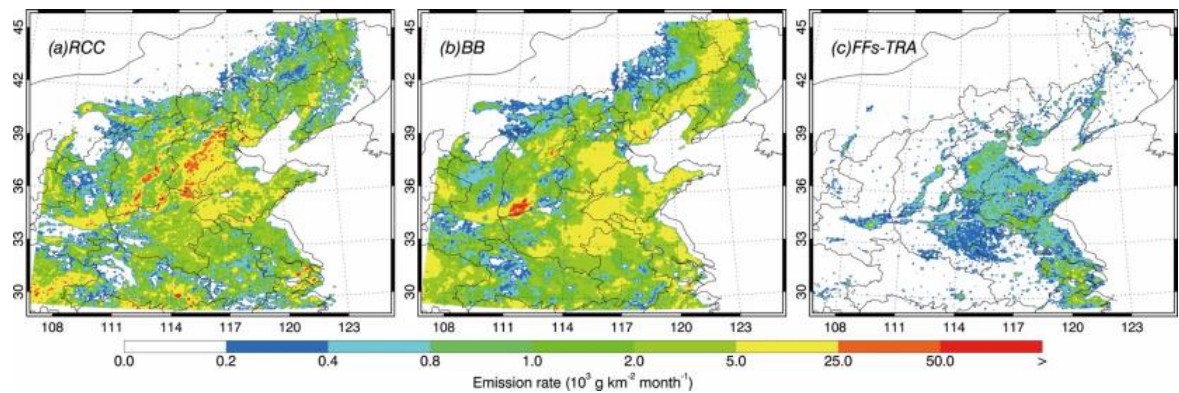


**Figure 2 Monthly BrC emissions burdens in January 2014 in NCP from RCC, BB and FFs-TRA.**

176         SOA has also shown light absorption in the atmosphere (Lin et al., 2014). Laboratory experiments have

revealed that most of the light-absorbing SOA is associated with aromatic SOA (Jacobson, 1999; Laskin et al.,
2015; Li et al., 2020) and the absorption from biogenic SOA in the field has been found to be negligible
(Washenfelder et al., 2015) . Therefore, here we assume aromatic derived SOA as secondary BrC in the model
following previous studies (Jo et al., 2016; Wang et al., 2018).

181         Moreover, it is worth noting that both primary and SOA light absorption were shown to be dynamic, where

BrC can be bleached when they undergo photodissociation (Forrister et al., 2015; Wong et al., 2019), or be darken
by cloud and fog processing of aerosols (Moise et al., 2015; Lin et al., 2017; Cheng et al., 2020). These processes
are not considered in this study yet. More detailed parameterization of the chemical aging of BrC are needed in
future BrC models.
**2.3.2 BrC optical properties**
The refractive indices of BrC as a function of wavelength are used for radiative transfer calculations. The
complex refractive index ($m = n + ik$) of aerosol components enables practical implementation in the model,
where $n$ is the real part primarily associated with the scattering efficiency, and $k$ is the imaginary part primarily
associated with the absorption efficiency (Bohren and Huffman, 1998). The real part of BrC refractive index is
the same of non-absorbing OA which is fairly constrained with reported values typically ranging from 1.5 to 1.7
(Saleh et al., 2014; Browne et al., 2019; Li et al., 2020). In this study, it follow the study by Li et al. (2011a) and
Wu et al. (2020). The imaginary part ($k$) of BrC refractive index exhibit strong wavelength dependence and the
values range over several orders of magnitude (Saleh et al., 2018; Sengupta et al., 2018). Limited studies on the
optical properties of BrC from fossil fuel combustions are reported at present. The average mass absorption
efficiency (MAE) of RCC at 365nm is ranging from 0.80 $m^2$ $g^{-1}$ to 2.47 $m^2$ $g^{-1}$ (Yan et al., 2017; Li et al., 2019;
Tang et al., 2020; Wang et al., 2020; Ni et al., 2021; Song et al., 2021; Wang et al., 2021). In this study, as shown
in Table 2, two sets of MAE are used for the sensitivity experiments of BrC. We choose a field optical
measurement of BrC from all sources made by Zhang et al., (2022b) as the high absorption case (HI-BRC-
ABS) .The optical properties of BB and FFs-TRA obtained in laboratory by Xie et al., (2017), as well as MAE of
RCC and secondary BrC obtained in laboratory by Ni et al, (2021) are adopted as the low absorption case (LOW-
BRC-ABS) in the study. The imaginary part of the two cases have shown wavelength dependent light-absorption
properties and the changes in anthropogenic emissions affect the optical properties of BrC. The imaginary part of
both two cases are interpolated to 11 wavelengths to match the aerosol radiation calculation of Goddard module
in WRF-Chem. The value of $k$ in this work is derived from the measured MAE using the following Eq.(6) (Liu
et al., 2013; Lu et al., 2015) as shown in Table2:
$k_{BrC,\lambda} = \frac{\rho \times \lambda \times MAE_\lambda}{4\pi}$                     (6)
Where $MAE_\lambda$ ($m^2$ $g^{-1}$) is the bulk mass absorption efficiency of BrC at the corresponding wavelength $\lambda$. $\rho$ (g
$cm^{-3}$) is the density of organic aerosols, which is assigned as 1.2 g $cm^{-3}$ (Turpin and Lim, 2001) in this study.

**Table 2 The refractive index of BrC used in the model**

| Aerosols | Wavelength (nm) | $k$ values for HI-BRC-ABS | $k$ values for LOW-BRC-ABS |
|---|---|---|---|
| BrC-RCC | 365 | - | 0.0320 |
| | 370 | 0.1890 | - |
| | 470 | 0.0608 | - |
| | 500 | - | 0.0020 |
| | 520 | 0.0272 | - |
| | 590 | 0.0173 | - |
| | 660 | 0.0081 | - |
| BrC-BB | 365 | - | 0.0300 |
| | 370 | 0.0587 | - |
| | 405 | - | 0.0016 |
| | 470 | 0.0219 | - |
| | 520 | 0.0120 | - |
| | 550 | - | 0.0026 |
| | 590 | 0.0092 | - |
| | 660 | 0.0046 | - |
| BrC-FFs-Tra | 365 | - | 0.0180 |
| | 370 | 0.0509 | - |
| | 405 | - | 0.0130 |
| | 470 | 0.0194 | - |
| | 520 | 0.0085 | - |
| | 550 | - | 0.0045 |
| | 590 | 0.0046 | - |
| | 660 | 0.0018 | - |
| BrC-SOA | 365 | - | 0.0049 |
| | 370 | 0.0251 | - |
| | 470 | 0.0166 | - |
| | 500 | - | 0.0007 |
| | 520 | 0.0114 | - |
| | 590 | 0.0107 | - |
| | 660 | 0.0063 | - |

'-' means not available

### 2.3.3 Shortwave direct radiative effect calculation and experimental design

The shortwave DRE calculations of BrC follow the method reported by Chen et al (2021) as shown in Eq. (7). The DRE of BrC is calculated by the difference between the net radiant flux with and without BrC, where the net radiant flux is the difference between the downward $(F_\downarrow)$ and upward radiant flux $(F_\uparrow)$.

$$DRE_{TOA} = (F^a_{\downarrow TOA} - F^a_{\uparrow TOA}) - (F^0_{\downarrow TOA} - F^0_{\uparrow TOA}) \tag{7}$$

Where $DRE_{TOA}$ represent the shortwave DRE at the top of the atmosphere (TOA). $F^a$ and $F^0$ are the radiant flux with and without BrC aerosols, respectively.

An adjoint methodology proposed by Zhao et al (2013) and Huang et al. (2015) has been used to diagnose the optical depth and DRE of BrC aerosols. Optical properties and radiative transfer of different sources BrC are calculated multiple times with one or a group of aerosol mass removed or without BrC absorption from each calculation following Eq. (8) and Eq. (9). In addition, the model also takes into account the reduced aerosol masses

along with the change in aerosol number concentration and size distribution.
$AOD_{[species\ i]} = AOD_{[all\ species]} - AOD_{[without\ species\ i/without\ species\ i\ absorption]}$ (8)
$DRE\ Forcing_{[species\ i]} = DRE\ Forcing_{[all\ species]} - DRE\ Forcing_{[without\ species\ i/without\ speices\ i\ absorption]}$ (9)
This method is more efficient than the traditional approach of running the model multiple times with the
exclusion of a specific aerosol component. It not only saves computational time but also provides a more accurate
estimation focused solely on the direct radiative effect of aerosols.
**3 Results and Discussions**
**3.1 Model performance**
Before evaluating the DRE of BrC, results from the standard simulation are used to validate the model
performance. Using available measurements, we first validate the spatial distribution and temporal variation of
air pollutants ($PM_{2.5}$, $O_3$, $NO_2$, $SO_2$) in the NCP, the temporal variation of downward shortwave flux at the surface
(SWDOWN) in Beijing, Tianjin, Zhengzhou, Hefei and Ji'nan, and the temporal variation of aerosol species (OA,
elemental carbon, ammonium, sulphate and nitrite) in Beijing and, Tianjin and of primary OA from BB, RCC,
motor vehicles and SOA in Beijing in January, 2014. Detailed data descriptions and quantitative statements of
model biases can be found in supplementary text S2.1 and S3. In general, the model simulates reasonably well the
air pollutants, SWDOWN, and aerosol species against measurements.
SSA determines the strength of aerosols in absorbing solar radiation. Here we conduct three sensitivity
experiments to evaluate the effect of BrC with different $k$ values on the simulated aerosol absorption. The first
experiment is the control simulation in which all organic aerosols are treated as purely scattering particles with no
absorption contribution of BrC, which is referred to as NOBRC. The hi-absorption scenario (HI-BRC-ABS) and
low-absorption scenario (LOW-BRC-ABS) characterize BrC light absorption by using the higher and lower
imaginary refractive index derived from Section 2.3.2, respectively. Figure 3 shows the comparisons of simulated
versus observed SSA at 440 nm ($SSA_{440}$) at Sun-sky radiometer Observation NETwork (SONET) sites in Beijing,
Songshan, Xi'an, Hefei, Nanjing in January 2014. Due to the influence of clouds, the observational data from
SONET are not continuous, resulting in a total of 237 valid data points are available for comparison. Moreover,
SSA retrieval typically have larger uncertainties at low AOD values (Dubovik et al., 2002). Therefore, we have
excluded the SSA data when AOD is less than 0.5, which has 206 valid points in each case. We find that the
inclusion of BrC in the model reduces the bias of simulated SSA. The HI-BRC-ABS case demonstrated a largest
improvement with the correlation coefficient increasing to 0.54, making it the best simulation in the study. It
suggests that stronger BrC absorption case, as prescribed in HI-BRC-ABS, better captures the aerosol optical
properties observed in northern China during winter. Consequently, the HI-BRC-ABS case can serve as the the
base simulation for further investigation of radiative effects of BrC in this study. Overall, the model tends to
underestimate $SSA_{440}$. The underestimation might be partly caused by the overestimation of absorbing aerosols
like BC or dust. Meanwhile, the uncertainties of the simulated SSA can be caused by other factors, such as mixing
state of aerosols, particle shape, wavelength, and mass ration of non-black carbon to BC (Liu et al., 2017; Jeong
et al., 2020).

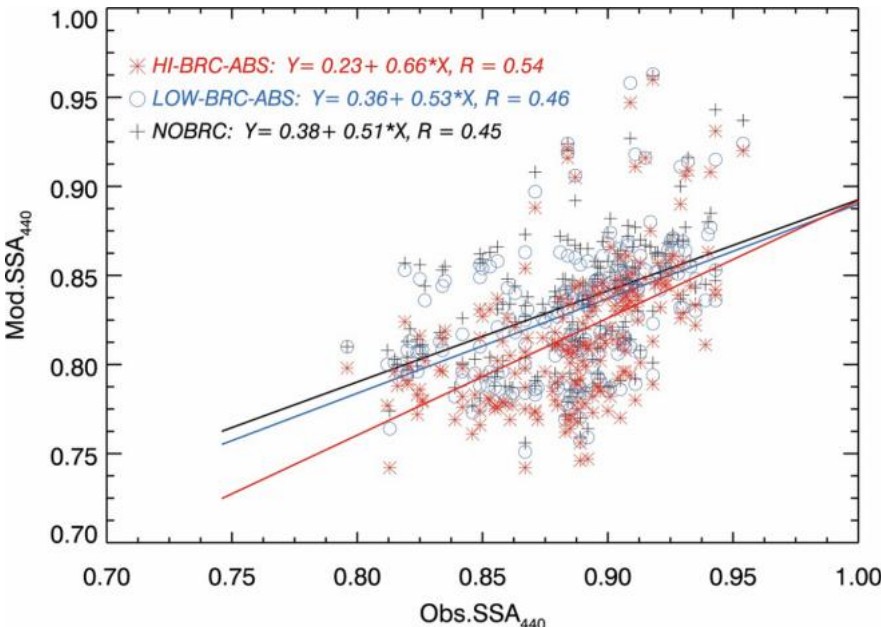


**Figure 3. Scatter plot and linear fitting of modelled and observed column integrated SSA at 440 nm in case HI-BRC-ABS (red), LOW-BRC-ABS (blue) and NOBRC (black) case.**

The daily AOD at 550 nm ($AOD_{550}$) from the dataset of Long-term Gap-free High-resolution Air Pollutant
(LGHAP), derived via tensor-flow-based multimodal data fusion method (Bai et al., 2022), is compared with the
simulation. This gap-free daily AOD dataset at 1 km resolution for 2000–2020 in China was generated by
integrating multimodal data from satellites, numerical models, and in situ measurements. Data gaps in Moderate
Resolution Imaging Spectroradiometer (MODIS) AOD are reconstructed through spatial pattern recognition and
statistical knowledge transfer. Validation against Aerosol Robotic Network (AERONET) observations showed
strong agreement, with an R of 0.91 and an RMSE of 0.21. Figure 4a and 4b shows the pattern comparison of the
monthly simulated and retrieved $AOD_{550}$. The model reproduces the retrieved AOD distribution in the NCP
reasonably. The monthly average simulated and retrieved $AOD_{550}$ is 0.45 and 0.48 on average in the NCP,
respectively. Figure 3c shows the scatter plot of the daily simulated and retrieved $AOD_{550}$ averaged in the NCP
during the simulation period. The simulated daily average $AOD_{550}$ correlates quite well with the retrieved value,
with a regression slope of 1.08 and correlation coefficient of 0.83. Generally, the retrieved and simulated AOD
increases with deterioration of the particulate pollution. Figure 5 provides the pattern comparison of the simulated
and Ozone Monitoring Instrument (OMI) retrieved AOD at 440 nm ($AOD_{440}$) averaged during the simulated
episode. OMI aboard NASA's Aura satellite offers global atmospheric measurements at a spatial resolution of
$0.25° \times 0.25°$, with Beijing's overpass occurring at approximately 13:45 local time. The averaged $AOD_{440}$ from the
model simulation at 14:00 local time shows generally agreement with the OMI retrieval. The average simulated
and retrieved $AOD_{440}$ is 0.51 and 0.53 in the NCP, respectively. Overall, the model generally performs well in
simulating the AOD distribution. It is worth noting that the simulated AOD is not only dependent on the column
aerosol content and composition, but is also substantially influenced by relative humidity (RH) which determines
the aerosol hygroscopic growth. Additionally, the satellite retrieved AOD is subject to contamination by the
presence of clouds, and considering the high occurrence frequency of clouds during haze days, the retrieved AOD
might be overestimated (Satheesh et al., 2010; Chand et al., 2012).

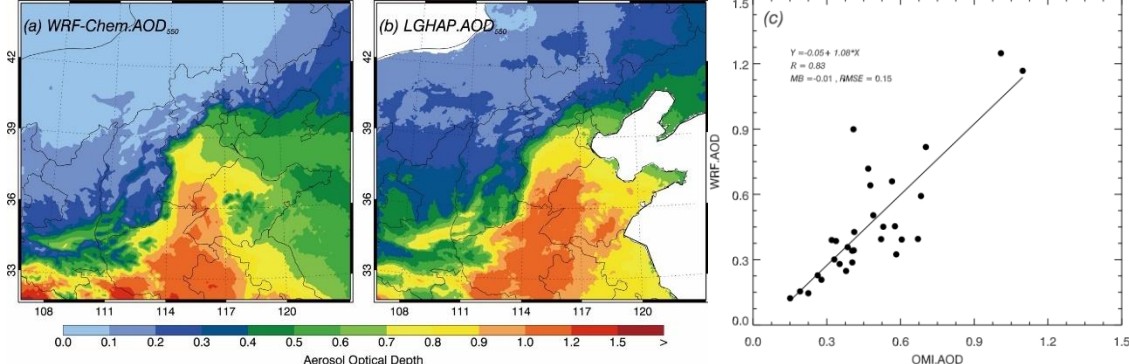


**Figure 4 (a) monthly simulated $AOD_{550}$ of WRF-Chem, (b) monthly retrieved $AOD_{550}$ of reanalysis dataset LGHAP,**
**and (c) scatter plot of the daily simulated and retrieved $AOD_{550}$ averaged in the NCP from 01 January to 30 January**
**2014**

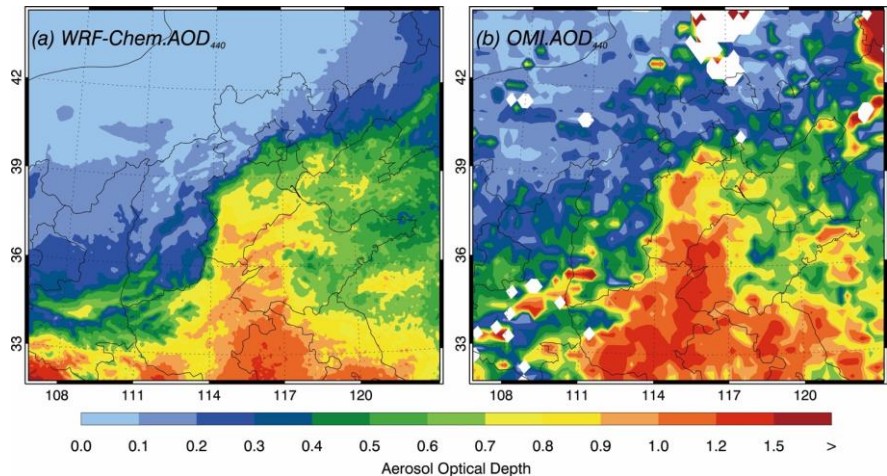


**Figure 5 (a) monthly simulated AOD$_{440}$ of WRF-Chem, (b) monthly retrieved AOD$_{440}$ of OMI in the NCP from 01 January to 30 January 2014**

**3.2 Surface mass concentrations of BrC in NCP**

The simulated distribution of average near-surface BrC concentrations and each source contribution in January 2014 is shown in Fig. 6a. In January, the monthly mean concentrations of BrC in the NCP vary from 0.05 $\mu g\ m^{-3}$ to 42.3 $\mu g\ m^{-3}$, with an average of 5.2 $\mu g\ m^{-3}$. The spatial distribution of near-surface BrC concentrations is like that of PM$_{2.5}$ in the NCP, with the highest concentration areas located in Hebei Province with an average concentration of 14.9 $\mu g\ m^{-3}$. The simulated BrC concentrations are higher than those reported by Zhu et al.(Bai et al., 2022; Zhu et al., 2021) (2021) in 2018, which is perhaps caused by the more severe particulate pollution in January 2014.

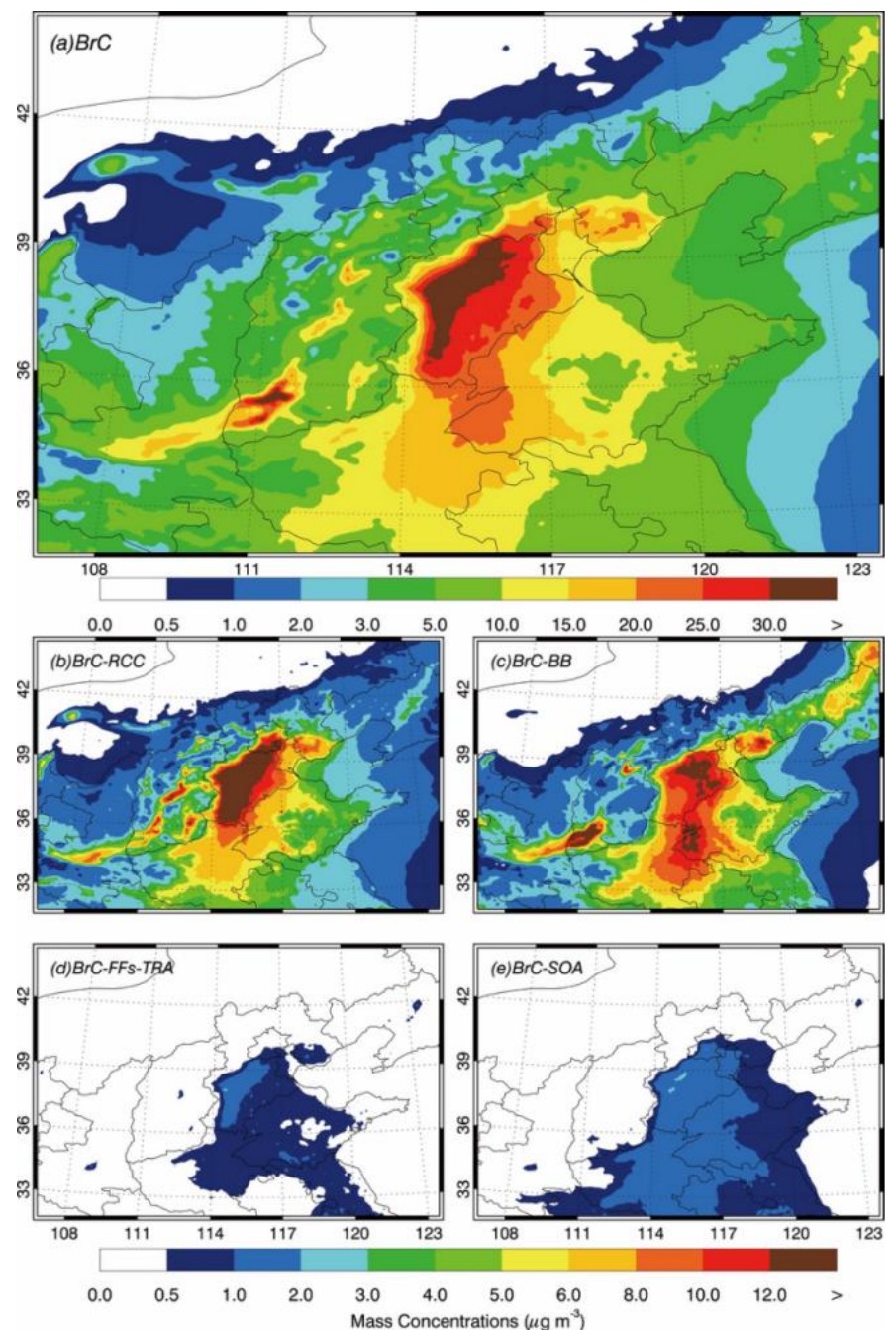

**Figure 6. Simulated mean surface concentrations of BrC (a) and contributions from RCC (b), BB (c), FFs-TRA (d) and secondary BrC (e) in January 2014 in NCP**

Figure 6b-e present the average near-surface BrC concentrations from different anthropogenic sources and secondary formation in January 2014 in the NCP. The BrC in the NCP predominantly originates from RCC and BB, with an average contribution of 2.3 and 2.4 $\mu g\ m^{-3}$ and a maximum of 33.8 and 31.6 $\mu g\ m^{-3}$, respectively. A relatively small proportion of BrC is related to FFs-TRA and secondary transformations with an average concentration of 0.2 $\mu g\ m^{-3}$ and 0.4 $\mu g\ m^{-3}$, respectively. The BrC from RCC accounts for 55.8% of total BrC

concentrations in the NCP, which is highest in Beijing, Hebei, and Tianjin, reaching 67.7%, 54.4% and 53.3%
respectively. The BrC from BB counts 36.9% of total BrC concentrations, with a contribution of about 40% in
most provinces of the NCP but only 26.5% in Beijing. This result shows that the RCC is one of the major sources
of BrC in NCP due to the wide use of coal for heating and cooking in winter with low combustion efficiency and
little emission control. The BrC emitted by RCC is mainly concentrated in the Beijing-Tianjin-Hebei (BTH) region
in January 2014, while the BrC emitted by BB is distributed in the whole NCP. The Fen-Wei plain exhibits notably
high contributions from BB, which is consistent with the emission distribution and with previous studies (Cao and
Cui, 2021; Zhang et al., 2021). The Fen-Wei Plain is one of the most densely populated and heavily polluted areas
in northern China where biomass is usually used for heating during winter.
**3.3 BrC absorption in the NCP**
BrC absorbs visible to near-ultraviolet light with its absorption capabilities extending prominently at shorter
wavelengths. Therefore, we calculate the AAOD (aerosol absorption optical depth) to evaluate the absorption
contribution of BrC versus bulk aerosols, each anthropogenic source and SOA versus BrC at 365 nm (Fig. 7), by
differentiating the AAOD between the model runs with and without the contribution of BrC. The average
contribution of BrC to the total AAOD of aerosols at 365 nm is 16.4% and the maximum is 39.5% in the NCP in
January 2014. The BrC to BC and OC ratios in surface air in the study is 2.2 and 0.31, respectively. They are
higher than the surface ratio used in the global models (Jo et al., 2016; Park et al., 2010), but lower than the ratio
of Feng et al., (2013). Although the concentrations of BrC is relatively high and the absorption of BrC in the
ultraviolet band is comparable to that of BC, the imaginary index of BrC (about 0.1) is still much lower than that
of BC (about 0.76). As a result, the light absorption contribution of BrC at 365 nm is not as significant during the
study period.
Furthermore, the light absorption properties of BrC during the winter season are predominantly attributed to
RCC, followed by BB, SOA, and FFs-TRA in the NCP. The average contribution of RCC and BB to the AAOD
of BrC is 59.3% and 26.3%, respectively. Although the concentration of BrC from RCC is comparable to that
from BB, the much higher light-absorbing property of the BrC from RCC makes it a dominant contributor to the
overall light absorption caused by BrC. Despite lower surface concentrations compared to primary BrC,
secondary BrC contributes significantly to AAOD of BrC, averaging ~10.0% with elevated contributions in
the sea and remote regions, which is likely due to the highly oxidized character of organic aerosols and its
chemical aging in aging air masses leading to the formation of BrC (Gouw et al., 2005; Kawamura et al.,

2005; Tsigaridis and Kanakidou, 2018). While AOD represents column-integrated concentrations, the secondary BrC to Primary BrC ratio increases from 8.9% at the surface to 12.0% of atmospheric burden. It reaches 14.3% at an altitude of 500m as shown in Figure S6, which could lead to a higher absorption contribution of secondary BrC (Wang et al., 2022b). Moreover, the observations indicate that a substantial SOA is water-soluble (Maria et al., 2003; Peng et al., 2021) which is treated as hygroscopic components in the model and its absorption could be magnified. The AAOD contribution of BrC from vehicles is the lowest, about 1.8% on average, but its contribution in southern Jiangsu and southeast Anhui ranges from 4% to 8%, higher than in other regions. This may suggest that motor vehicle emissions account for a significant proportion of pollution in this area.

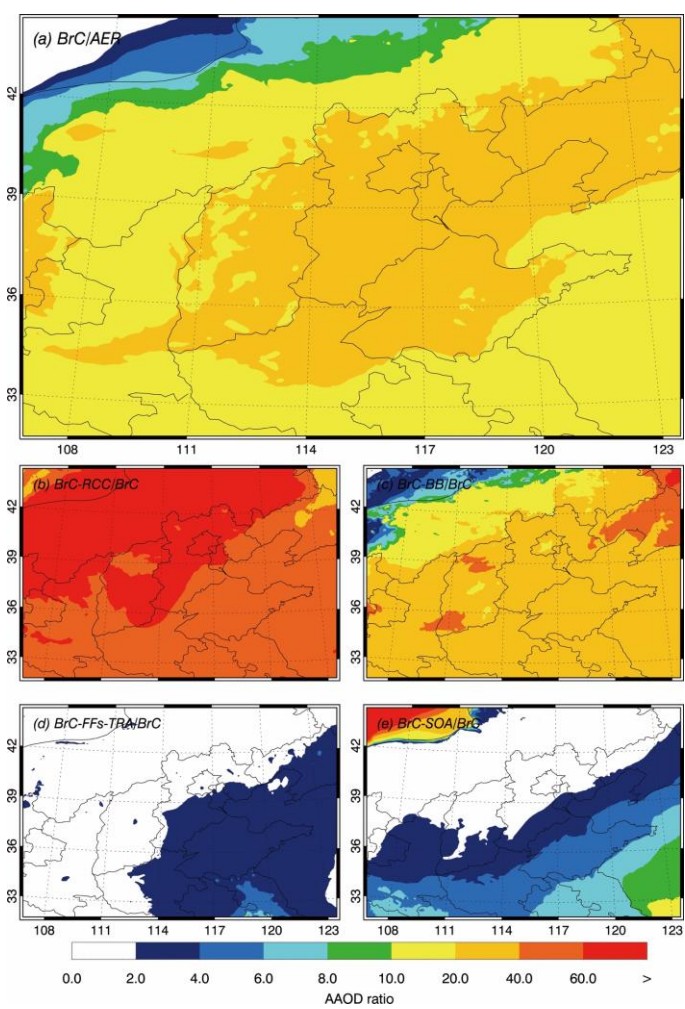

**Figure 7 Simulated monthly BrC AAOD versus aerosol at 365nm (a) and each anthropogenic source of RCC (b), BB(c), FFs-TRA (d) and secondary BrC (d) AAOD versus BrC in NCP**

**3.4 Direct radiative effect of BrC**

353       Figure. 8a shows the distribution of the average $DRE_{TOA}$ caused by BrC at the top of the atmosphere during

the episode. The $DRE_{TOA}$ of BrC in the NCP is -0.09 W m$^{-2}$ on average, with a maximum of +0.46 W m$^{-2}$ and a
minimum of -2.74 W m$^{-2}$. Compared to the average $DRE_{TOA}$ of BC +3.9 W m$^{-2}$ and a maximum of +21.6 W m$^{-2}$,
the average $DRE_{TOA}$ of BrC in the NCP is close to zero. However, in terms of the spatial distribution, the $DRE_{TOA}$
of total BrC in the NCP is predominantly negative, especially in those areas with high BrC concentrations
including BTH area and Fen-wei Plain. The largest negative $DRE_{TOA}$ is in Shanxi province, where the highest
contributor to BrC is BB. The overall scattering effect of BrC is greater than its absorption effect, so that BrC
have a net cooling effect. The solar irradiance in the UV band contributes only 10% of the total solar irradiation
and the imaginary refractive indices of BrC are much lower than those strongly absorbing BC, especially in the
visible band. On the other hand, although this leads to a small heating effect by BrC, the increased $DRE_{TOA}$ induced
by BrC which is usually considered as its scattering effect, is up to an average of +0.40 W m$^{-2}$ and a maximum
of +1.83 W m$^{-2}$ as shown in Fig. 8b. The $DRE_{TOA}$ of OA without BrC is -2.00 W m$^{-2}$ over the NCP (Fig. 8d) and
is increased to -1.60 W m$^{-2}$ (Fig. 8c) after considering BrC. Consequently, the cooling effect of OA is reduced by
BrC by an average of 28.0%. So far, almost all estimates of the radiation effects of BrC have been based on a
global basis. The study of Wang et al. (2014) reported a global $DRE_{TOA}$ of BrC of -0.02 W m$^{-2}$, resulting in a
$DRE_{TOA}$ increase of +0.07 W m$^{-2}$. Brown et al. (2018) have also reported a global annual increased $DRE_{TOA}$ of
+0.13 W m$^{-2}$, with the maximum forcing ($\sim$+1.75 W m$^{-2}$) occurring at the west coast of southern Africa. BrC
reduces the cooling effect caused by organic aerosols by approximately 16% (Jo et al., 2016). However, all these
estimates only considered BB, biofuels or SOA and the significance of BrC in the estimates for these studies
typically based on the assumption that OA primarily scatters sun light. Our results indicate that the solar radiation
changes caused by BrC in NCP is notable. The absorption effect of BrC should be considered in climate models
to accurately assess the aerosol impact on atmospheric heating and climate change.

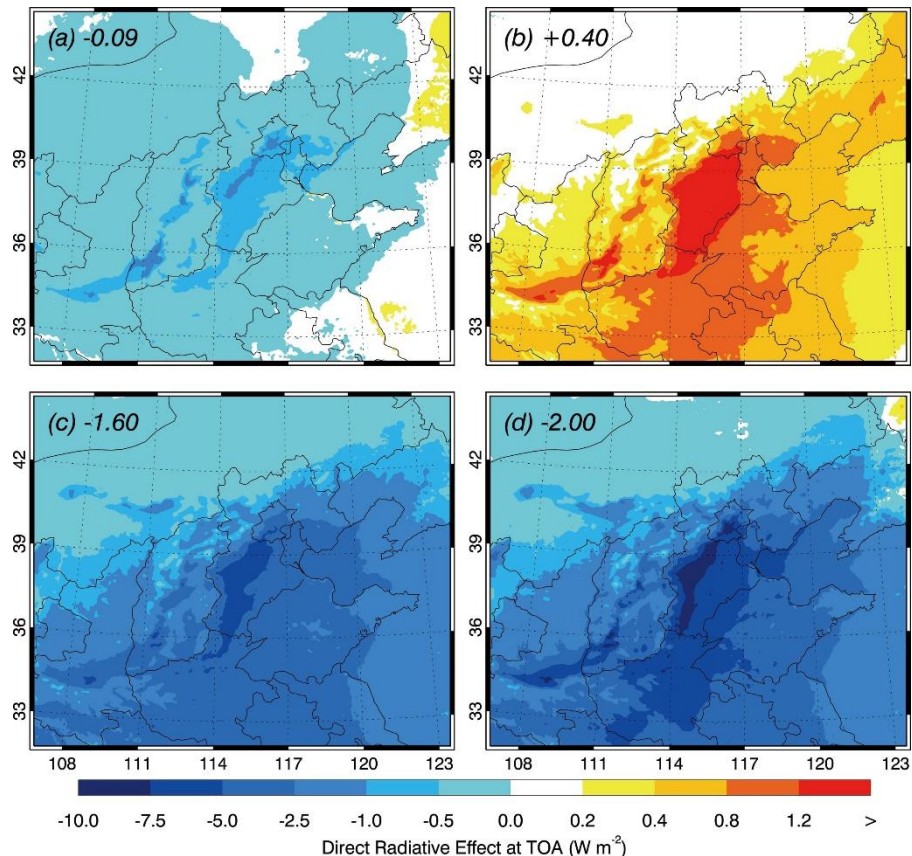

**Figure 8,** Estimated DRE$_{TOA}$ of BrC (a) and DRE$_{TOA}$ increase of OA owing to the absorption of BrC (b). The DRE$_{TOA}$ of total OA with absorbing BrC (c) and DRE$_{TOA}$ of total OA with BrC is assumed to be scattering (d). The averages of DRE are shown in the upper left of each panel.

Figure 9a-d shows the estimated DRE$_{TOA}$ of BrC from RCC, BB, FFs-TRA and secondary BrC in the NCP during the episode. Similar to the contribution of BrC sources to the AAOD at 365 nm, the most important source contributing to DRE$_{TOA}$ of BrC is RCC (+0.26 W m$^{-2}$), followed by BB (+0.11 W m$^{-2}$), secondary BrC (+0.02 W m$^{-2}$), and FFs-TRA (+0.01 W m$^{-2}$) in the NCP, as shown in Fig. 9. In addition, the DRE$_{TOA}$ of BrC from various sources exhibits distinct spatial distribution characteristics in the NCP. The highest DRE$_{TOA}$ of BrC from fossil sources, which include RCC and FFs-TRA, are predominantly concentrated in Hebei. The highest positive DRE$_{TOA}$ of BrC from BB is found in Fen-Wei Plain. Meanwhile, the secondary BrC contributes most to the DRE$_{TOA}$ in the south part of the NCP. The persistent high values over southern China might stem from the model's representation of secondary BrC as hygroscopic components (Peng et al., 2021), whose light-absorbing capacity is amplified in the region with high ambient humidity.

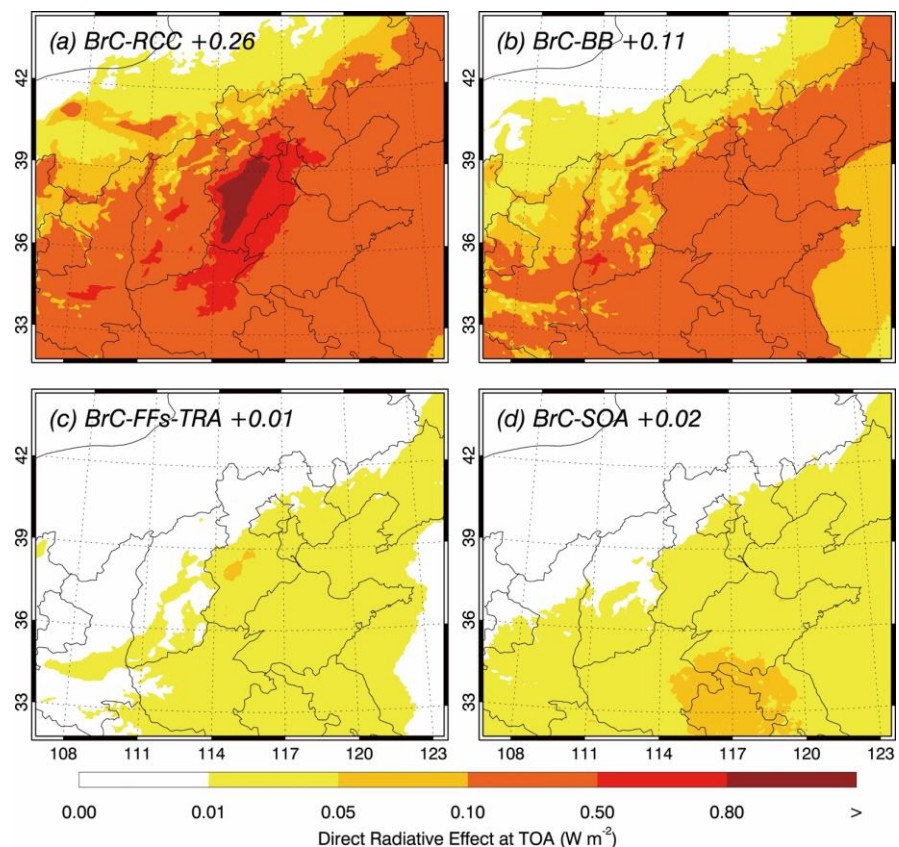

**Figure 9. The estimation DRE$_{TOA}$ of BrC from each anthropogenic source RCC(a), BB(b), FFs-TRA(c) and secondary BrC(d) in NCP in January 2014**

**4 Conclusions**

A source-explicit BrC simulation in January 2014 in the NCP is conducted by using the WRF-Chem model. We define the BrC based on varying proportions of RCC, BB, FFs-TRA and SOA sources, assigning each a distinct imaginary refractive index to represent their differing light absorption characteristics. Model simulations are evaluated with various data sets. Besides the well reproduction of temporal and spatial variations of aerosol components and SWDOWN in the model, AOD and SSA are used to evaluate the aerosol optical properties.

Near-surface mass concentrations of simulated BrC in the NCP range from 0.05 to 42.3 μg m$^{-3}$ with an average of 5.2 μg m$^{-3}$, which is mainly attributed to RCC and BB, especially in the BTH region and Fen-Wei Plain. Estimation of the BrC contribution to AAOD shows that the BrC accounts for an average of 16.4% and up to 39.5% of the total aerosol absorption at 365 nm. The largest contributor to the absorption of BrC is RCC derived BrC, reaching 59.3%. BrC generally has a net cooling effect in the NCP if we consider both the absorption and scattering properties, with DRE$_{TOA}$ of BrC of around -0.09 W m$^{-2}$ on average and ranging between -2.74 W m$^{-2}$ and +0.46

W m$^{-2}$. However, the absorption of BrC increases the DRE$_{TOA}$ of OA by 28.0% with an average of +0.40 W m$^{-2}$
and a maximum of +1.83 W m$^{-2}$. The average increase in DRE$_{TOA}$ of BrC from RCC, BB, secondary formation,
and FFs-TRA is +0.26 W m$^{-2}$, +0.11 W m$^{-2}$, +0.02 W m$^{-2}$, +0.01 W m$^{-2}$, respectively. Our results indicate that BrC
derived from RCC may have significant implications for regions relying heavily on coal as their primary energy
source, such as northern China. Climate models should not only incorporate the absorption of BrC but also account
for residential coal burning as a potentially important BrC emission source. Additionally, although we conducted
simulations with a relatively conservative secondary BrC, the impact of secondary BrC on radiative processes
should not be overlooked. More field observation and model experiments should be carried out in the future for
better understanding of its role in atmospheric radiation balance.
It should be noted that China has started to switch from coal to cleaner and more efficient energy such as
natural gas or liquid petroleum gas in recent years. According to the latest report of National Bureau of Statistics
of China, the total coal consumption for residential use was 55.5 Gg in 2022 (https://data.stats.gov.cn)with a
40.3% decrease compared to 2014. Therefore, our diagnosis of the sources of BrC and their radiative effects is
specifically targeted at the winter season in 2014. Moreover, future simulations should strengthen the
parameterization for the evolution of BrC, such as bleaching or darkening processes.
The simulation of BrC in climate models is fraught with uncertainties due to its diverse sources which result
in a wide range of optical properties. The absorption characteristics of BrC can change significantly as it undergoes
atmospheric aging, impacting its radiative forcing estimates. Additionally, interactions of BrC with other
atmospheric particles and its effects on cloud microphysics and albedo introduce further complexities in modeling
its climate impact. These uncertainties necessitate enhanced observational data and better integration of BrC
properties in climate models to improve the accuracy of climate predictions and assessments. Although the
simulations have been evaluated with extensive aerosol mass and optical measurements, more field measurements
and lab experiments are needed, especially for the inventory development of BrC, the vertical profiles from
aircrafts, and physicochemical properties which would be useful for further evaluating and improving model
performance.
**Author contributions**
Guohui Li and Xuexi Tie designed the study. Jiamao Zhou and Guohui Li wrote the paper. Jiamao Zhou,
Jiarui Wu, Xi Li, Ruonan Wang performed the model simulations. Xiaoli Su and Jiang Qian collected satellites
and ground-based observation data. Ting Zhang, Wenting Dai and Junji Cao performed field observation and
laboratory analysis. Imad EI Haddad and Andre S.H. Prevot provided the data of OA components in Beijing,
kindly reviewed the language writing of the manuscript and provided some supplementary suggestions for the
paper. All authors reviewed and commented on the paper.
**Acknowledgments**
This work was financially supported by the project of Young Scientists Fund of the National Natural Science
Foundation of China (42107127).
**Data availability**
The Chinese Ecosystem Research Network (CERN) provided the radiation observation data. The $AOD_{550}$
data is supported from "National Earth System Science Data Center (https://www.geodata.cn)" and $AOD_{440}$
provided by OMI Science Team (https://www.earthdata.nasa.gov/learn/find-data/near-real-time/omi). The SSA
data is supported by Sun-sky Radiometer Observation NETwork (http://www.sonet.ac.cn) and the historic profiles
of the observed ambient air pollutants provided by Ministry of Ecology and Environment of China
(https://www.aqistudy.cn/).
**Competing interests**
The authors declare that they have no conflict of interest.

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

Contributions of residential coal combustion to the air quality in Beijing–Tianjin–Hebei (BTH), China: a case
study, Atmos. Chem. Phys., 18, 10675–10691, https://doi.org/10.5194/acp-18-10675-2018, 2018.
Lin, G., Penner, J. E., Flanner, M. G., Sillman, S., Xu, L., and Zhou, C.: Radiative forcing of organic aerosol in
the atmosphere and on snow: Effects of SOA and brown carbon, J. Geophys. Res., 119, 7453–7476,
https://doi.org/10.1002/2013JD021186, 2014.
Lin, P., Bluvshtein, N., Rudich, Y., Nizkorodov, S. A., Laskin, J., and Laskin, A.: Molecular Chemistry of
Atmospheric Brown Carbon Inferred from a Nationwide Biomass Burning Event, Environmental science &
technology, 51, 11561–11570, https://doi.org/10.1021/acs.est.7b02276, 2017.
Liu, D., Whitehead, J., Alfarra, M. R., Reyes-Villegas, E., Spracklen, D. V., Reddington, C. L., Kong, S., Williams,
P. I., Ting, Y.-C., Haslett, S., Taylor, J. W., Flynn, M. J., Morgan, W. T., McFiggans, G., Coe, H., and Allan, J. D.:
Black-carbon absorption enhancement in the atmosphere determined by particle mixing state, Nature Geosci, 10,
184–188, https://doi.org/10.1038/ngeo2901, 2017.
Liu, J., Bergin, M., Guo, H., King, L., Kotra, N., Edgerton, E., and Weber, R. J.: Size-resolved measurements of
brown carbon in water and methanol extracts and estimates of their contribution to ambient fine-particle light
absorption, Atmos. Chem. Phys., 13, 12389–12404, https://doi.org/10.5194/acp-13-12389-2013, 2013.
Lu, Z., Streets, D. G., Winijkul, E., Yan, F., Chen, Y., Bond, T. C., Feng, Y., Dubey, M. K., Liu, S., Pinto, J. P., and
Carmichael, G. R.: Light absorption properties and radiative effects of primary organic aerosol emissions,
Environmental science & technology, 49, 4868–4877, https://doi.org/10.1021/acs.est.5b00211, 2015.
Ma, Y., Jin, Y., Zhang, M., Gong, W., Hong, J., Jin, S., Shi, Y., Zhang, Y., and Liu, B.: Aerosol optical properties
of haze episodes in eastern China based on remote-sensing observations and WRF-Chem simulations, The Science
of the total environment, 757, 143784, https://doi.org/10.1016/j.scitotenv.2020.143784, 2021.
Maria, S. F., Russell, L. M., Turpin, B. J., Porcja, R. J., Campos, T. L., Weber, R. J., and Huebert, B. J.: Source
signatures of carbon monoxide and organic functional groups in Asian Pacific Regional Aerosol Characterization
Experiment (ACE-Asia) submicron aerosol types, J. Geophys. Res., 108, 295,
https://doi.org/10.1029/2003JD003703, 2003.
Mo, Y., Li, J., Cheng, Z., Zhong, G., Zhu, S., Tian, C., Chen, Y., and Zhang, G.: Dual Carbon Isotope-Based
Source Apportionment and Light Absorption Properties of Water-Soluble Organic Carbon in PM 2.5 Over China,
J. Geophys. Res., 126, 27805, https://doi.org/10.1029/2020JD033920, 2021.

Moise, T., Flores, J. M., and Rudich, Y.: Optical properties of secondary organic aerosols and their changes by chemical processes, Chemical reviews, 115, 4400–4439, https://doi.org/10.1021/cr5005259, 2015.

Mukai, H. and Ambe, Y.: Characterization of a humic acid-like brown substance in airborne particulate matter and tentative identification of its origin, Atmospheric Environment (1967), 20, 813–819, https://doi.org/10.1016/0004-6981(86)90265-9, 1986.

Nenes, A., Pandis, S. N., and Pilinis Christodoulos: ISORROPIA: A New Thermodynamic Equilibrium Model for Multiphase Multicomponent Inorganic Aerosols, Aquatic Geochemistry, 4, 123–152, 1998.

Ni, H., Huang, R.-j., Pieber, S. M., Corbin, J. C., Stefenelli, G., Pospisilova, V., Klein, F., Gysel-Beer, M., Yang, L., Baltensperger, U., Haddad, I. E., Slowik, J. G., Cao, J., Prévôt, A. S. H., and Dusek, U.: Brown Carbon in Primary and Aged Coal Combustion Emission, Environmental science & technology, 55, 5701–5710, https://doi.org/10.1021/acs.est.0c08084, 2021.

Park, R. J., Kim, M. J., Jeong, J. I., Youn, D., and Kim, S.: A contribution of brown carbon aerosol to the aerosol light absorption and its radiative forcing in East Asia, Atmospheric Environment, 44, 1414–1421, https://doi.org/10.1016/j.atmosenv.2010.01.042, 2010.

Peng, C., Razafindrambinina, P. N., Malek, K. A., Chen, L., Wang, W., Huang, R.-j., Zhang, Y., Ding, X., Ge, M., Wang, X., Asa-Awuku, A. A., and Tang, M.: Interactions of organosulfates with water vapor under sub- and supersaturated conditions, Atmos. Chem. Phys., 21, 7135–7148, https://doi.org/10.5194/acp-21-7135-2021, 2021.

Pokhrel, R. P., Beamesderfer, E. R., Wagner, N. L., Langridge, J. M., Lack, D. A., Jayarathne, T., Stone, E. A., Stockwell, C. E., Yokelson, R. J., and Murphy, S. M.: Relative importance of black carbon, brown carbon, and absorption enhancement from clear coatings in biomass burning emissions, Atmos. Chem. Phys., 17, 5063–5078, https://doi.org/10.5194/acp-17-5063-2017, 2017.

Saleh, R.: From Measurements to Models: Toward Accurate Representation of Brown Carbon in Climate Calculations, Curr Pollution Rep, 6, 90–104, https://doi.org/10.1007/s40726-020-00139-3, 2020.

Saleh, R., Cheng, Z., and Atwi, K.: The Brown–Black Continuum of Light-Absorbing Combustion Aerosols, Environ. Sci. Technol. Lett., 5, 508–513, https://doi.org/10.1021/acs.estlett.8b00305, 2018.

Saleh, R., Robinson, E. S., Tkacik, D. S., Ahern, A. T., Liu, S., Aiken, A. C., Sullivan, R. C., Presto, A. A., Dubey, M. K., Yokelson, R. J., Donahue, N. M., and Robinson, A. L.: Brownness of organics in aerosols from biomass burning linked to their black carbon content, Nature Geosci, 7, 647–650, https://doi.org/10.1038/ngeo2220, 2014.

Shapiro, E. L., Szprengiel, J., Sareen, N., Jen, C. N., Giordano, M. R., and McNeill, V. F.: Light-absorbing

secondary organic material formed by glyoxal in aqueous aerosol mimics, Atmos. Chem. Phys., 9, 2289–2300, https://doi.org/10.5194/acp-9-2289-2009, 2009.

Song, S., Wang, Y., Wang, Y., Wang, T., and Tan, H.: The characteristics of particulate matter and optical properties of Brown carbon in air lean condition related to residential coal combustion, Powder Technology, 379, 505–514, https://doi.org/10.1016/j.powtec.2020.10.082, 2021.

Stokes, R. H. and Robinson, R. A.: Interactions in Aqueous Nonelectrolyte Solutions. I. Solute-Solvent Equilibria, J. Phys. Chem., 70, 2126–2131, https://doi.org/10.1021/j100879a010, 1966.

Sun, J., Zhang, Y., Zhi, G., Hitzenberger, R., Jin, W., Chen, Y., Wang, L., Tian, C., Li, Z., Chen, R., Xiao, W., Cheng, Y., Yang, W., Yao, L., Cao, Y., Huang, D., Qiu, Y., Xu, J., Xia, X., Yang, X., Zhang, X., Zong, Z., Song, Y., and Wu, C.: Brown carbon's emission factors and optical characteristics in household biomass burning: Developing a novel algorithm for estimating the contribution of brown carbon, Atmos. Chem. Phys., 21, 2329–2341, https://doi.org/10.5194/acp-21-2329-2021, 2021.

Sun, J., Zhi, G., Hitzenberger, R., Chen, Y., Tian, C., Zhang, Y., Feng, Y., Cheng, M., Zhang, Y., Cai, J., Chen, F., Qiu, Y., Jiang, Z., Li, J., Zhang, G., and Mo, Y.: Emission factors and light absorption properties of brown carbon from household coal combustion in China, Atmos. Chem. Phys., 17, 4769–4780, https://doi.org/10.5194/acp-17-4769-2017, 2017.

Tang, J., Li, J., Su, T., Han, Y., Mo, Y., Jiang, H., Cui, M., Jiang, B., Chen, Y., Tang, J., Song, J., Peng, P.'a., and Zhang, G.: Molecular compositions and optical properties of dissolved brown carbon in biomass burning, coal combustion, and vehicle emission aerosols illuminated by excitation–emission matrix spectroscopy and Fourier transform ion cyclotron resonance mass spectrometry analysis, Atmos. Chem. Phys., 20, 2513–2532, https://doi.org/10.5194/acp-20-2513-2020, 2020.

Tian, J., Wang, Q., Ni, H., Wang, M., Zhou, Y., Han, Y., Shen, Z., Pongpiachan, S., Zhang, N., Zhao, Z., Zhang, Q., Zhang, Y., Long, X., and Cao, J.: Emission Characteristics of Primary Brown Carbon Absorption From Biomass and Coal Burning: Development of an Optical Emission Inventory for China, J. Geophys. Res. Atmos., 15, 27,805, https://doi.org/10.1029/2018JD029352, 2019.

Tsigaridis, K. and Kanakidou, M.: The Present and Future of Secondary Organic Aerosol Direct Forcing on Climate, Curr Clim Change Rep, 4, 84–98, https://doi.org/10.1007/s40641-018-0092-3, 2018.

Tuccella, P., Curci, G., Pitari, G., Lee, S., and Jo, D. S.: Direct Radiative Effect of Absorbing Aerosols: Sensitivity to Mixing State, Brown Carbon, and Soil Dust Refractive Index and Shape, J. Geophys. Res. Atmos., 125, 317,

https://doi.org/10.1029/2019JD030967, 2020.

Turpin, B. J. and Lim, H.-J.: Species Contributions to PM2.5 Mass Concentrations: Revisiting Common Assumptions for Estimating Organic Mass, Aerosol Science and Technology, 35, 602–610, https://doi.org/10.1080/02786820152051454, 2001.

Wang, J., Ye, J., Zhang, Q., Zhao, J., Wu, Y., Li, J., Liu, D., Li, W., Zhang, Y., Wu, C., Xie, C., Qin, Y., Lei, Y., Huang, X., Guo, J., Liu, P., Fu, P., Li, Y., Lee, H. C., Choi, H., Zhang, J., Liao, H., Chen, M., Sun, Y., Ge, X., Martin, S. T., and Jacob, D. J.: Aqueous production of secondary organic aerosol from fossil-fuel emissions in winter Beijing haze, Proceedings of the National Academy of Sciences of the United States of America, 118, https://doi.org/10.1073/pnas.2022179118, 2021.

Wang, Q., Zhou, Y., Ma, N., Zhu, Y., Zhao, X., Zhu, S., Tao, J., Hong, J., Wu, W., Cheng, Y., and Su, H.: Review of Brown Carbon Aerosols in China: Pollution Level, Optical Properties, and Emissions, J. Geophys. Res., 127, 455, 2022a.

Wang, Q., Wang, L., Gong, C., Li, M., Xin, J., Tang, G., Sun, Y., Gao, J., Wang, Y., Wu, S., Kang, Y., Yang, Y., Li, T., Liu, J., and Wang, Y.: Vertical evolution of black and brown carbon during pollution events over North China Plain, The Science of the total environment, 806, 150950, https://doi.org/10.1016/j.scitotenv.2021.150950, 2022b.

Wang, X., Heald, C. L., Ridley, D. A., Schwarz, J. P., Spackman, J. R., Perring, A. E., Coe, H., Liu, D., and Clarke, A. D.: Exploiting simultaneous observational constraints on mass and absorption to estimate the global direct radiative forcing of black carbon and brown carbon, Atmos. Chem. Phys., 14, 10989–11010, https://doi.org/10.5194/acp-14-10989-2014, 2014.

Wang, X., Heald, C. L., Liu, J., Weber, R. J., Campuzano-Jost, P., Jimenez, J. L., Schwarz, J. P., and Perring, A. E.: Exploring the observational constraints on the simulation of brown carbon, Atmos. Chem. Phys., 18, 635–653, https://doi.org/10.5194/acp-18-635-2018, 2018.

Wang, Y., Wang, Y., Song, S., Wang, T., Li, D., and Tan, H.: Effects of coal types and combustion conditions on carbonaceous aerosols in flue gas and their light absorption properties, Fuel, 277, 118148, https://doi.org/10.1016/j.fuel.2020.118148, 2020.

Washenfelder, R. A., Attwood, A. R., Brock, C. A., Guo, H., Xu, L., Weber, R. J., Ng, N. L., Allen, H. M., Ayres, B. R., Baumann, K., Cohen, R. C., Draper, D. C., Duffey, K. C., Edgerton, E., Fry, J. L., Hu, W. W., Jimenez, J. L., Palm, B. B., Romer, P., Stone, E. A., Wooldridge, P. J., and Brown, S. S.: Biomass burning dominates brown carbon absorption in the rural southeastern United States, Geophys. Res. Lett., 42, 653–664,

https://doi.org/10.1002/2014GL062444, 2015.
Wong, J. P. S., Tsagkaraki, M., Tsiodra, I., Mihalopoulos, N., Violaki, K., Kanakidou, M., Sciare, J., Nenes, A.,
and Weber, R. J.: Atmospheric evolution of molecular-weight-separated brown carbon from biomass burning,
Atmos. Chem. Phys., 19, 7319–7334, https://doi.org/10.5194/acp-19-7319-2019, 2019.
Wu, J., Bei, N., Hu, B., Liu, S., Wang, Y., Shen, Z., Li, X., Liu, L., Wang, R., Liu, Z., Cao, J., Tie, X., Molina, L.
T., and Li, G.: Aerosol-photolysis interaction reduces particulate matter during wintertime haze events,
Proceedings of the National Academy of Sciences of the United States of America, 117, 9755–9761,
https://doi.org/10.1073/pnas.1916775117, 2020.
Xie, C., Xu, W., Wang, J., Wang, Q., Liu, D., Tang, G., Chen, P., Du, W., Zhao, J., Zhang, Y., Zhou, W., Han, T.,
Bian, Q., Li, J., Fu, P., Wang, Z., Ge, X., Allan, J., Coe, H., and Sun, Y.: Vertical characterization of aerosol optical
properties and brown carbon in winter in urban Beijing, China, Atmos. Chem. Phys., 19, 165–179,
https://doi.org/10.5194/acp-19-165-2019, 2019.
Xie, M., Hays, M. D., and Holder, A. L.: Light-absorbing organic carbon from prescribed and laboratory biomass
burning and gasoline vehicle emissions, Scientific reports, 7, 7318, https://doi.org/10.1038/s41598-017-06981-8,

724 2017.

Xu, L., Lin, G., Liu, X., Wu, C., Wu, Y., and Lou, S.: Constraining Light Absorption of Brown Carbon in China
and Implications for Aerosol Direct Radiative Effect, Geophys. Res. Lett., 51, 455,
https://doi.org/10.1029/2024GL109861, 2024.
Yan, C., Zheng, M., Bosch, C., Andersson, A., Desyaterik, Y., Sullivan, A. P., Collett, J. L., Zhao, B., Wang, S.,
He, K., and Gustafsson, Ö.: Important fossil source contribution to brown carbon in Beijing during winter,
Scientific reports, 7, 43182, https://doi.org/10.1038/srep43182, 2017.
Yan, J., Wang, X., Gong, P., Wang, C., and Cong, Z.: Review of brown carbon aerosols: Recent progress and
perspectives, The Science of the total environment, 634, 1475–1485,
https://doi.org/10.1016/j.scitotenv.2018.04.083, 2018.
Yang, M., Howell, S. G., Zhuang, J., and Huebert, B. J.: Attribution of aerosol light absorption to black carbon,
brown carbon, and dust in China – interpretations of atmospheric measurements during EAST-AIRE, Atmos.
Chem. Phys., 9, 2035–2050, https://doi.org/10.5194/acp-9-2035-2009, 2009.
Zhang, A., Wang, Y., Zhang, Y., Weber, R. J., Song, Y., Ke, Z., and Zou, Y.: Modeling the global radiative effect
of brown carbon: A potentially larger heating source in the tropical free troposphere than black carbon, Atmos.

Chem. Phys., 20, 1901–1920, https://doi.org/10.5194/acp-20-1901-2020, 2020.

Zhang, Q., Streets, D. G., Carmichael, G. R., He, K. B., Huo, H., Kannari, A., Klimont, Z., Park, I. S., Reddy, S., Fu, J. S., Chen, D., Duan, L., Lei, Y., Wang, L. T., and Yao, Z. L.: Asian emissions in 2006 for the NASA INTEX-B mission, Atmos. Chem. Phys., 9, 5131–5153, https://doi.org/10.5194/acp-9-5131-2009, 2009.

Zhang, Q., Li, Z., Shen, Z., Zhang, T., Zhang, Y., Sun, J., Zeng, Y., Xu, H., Wang, Q., Hang Ho, S. S., and Cao, J.: Source profiles of molecular structure and light absorption of PM2.5 brown carbon from residential coal combustion emission in Northwestern China, Environmental pollution (Barking, Essex 1987), 299, 118866, https://doi.org/10.1016/j.envpol.2022.118866, 2022a.

Zhang, W., Wang, W., Li, J., Ma, S., Lian, C., Li, K., Shi, B., Liu, M., Li, Y., Wang, Q., Sun, Y., Tong, S., and Ge, M.: Light absorption properties and potential sources of brown carbon in Fenwei Plain during winter 2018-2019, Journal of environmental sciences (China), 102, 53–63, https://doi.org/10.1016/j.jes.2020.09.007, 2021.

Zhang, Y., Wang, Q., Tian, J., Li, Y., Liu, H., Ran, W., Han, Y., Prévôt, A. S.H., and Cao, J.: Impact of COVID-19 lockdown on the optical properties and radiative effects of urban brown carbon aerosol, Geoscience Frontiers, 13, 101320, https://doi.org/10.1016/j.gsf.2021.101320, 2022b.

Zhao, C., Ruby Leung, L., Easter, R., Hand, J., and Avise, J.: Characterization of speciated aerosol direct radiative forcing over California, J. Geophys. Res., 118, 2372–2388, https://doi.org/10.1029/2012JD018364, 2013.

Zhu, Y., Wang, Q., Yang, X., Yang, N., and Wang, X.: Modeling Investigation of Brown Carbon Aerosol and Its Light Absorption in China, Atmosphere, 12, 892, https://doi.org/10.3390/atmos12070892, 2021.