# Peer review of "Source-explicit estimation of brown carbon in the polluted atmosphere over North China Plain: implications for"

_EGUsphere, 2024_

## Author Comment (AC1)

**Reply to Anonymous Referee #1**

We thank the reviewer for the careful reading of the manuscript and helpful comments. We have revised the manuscript following the suggestion, as described below.

**1 Comment:** L131-135: A major methodological concern is the arbitrary assumption that 10% of total SOA contributes to BrC, which lacks scientific justification. While the authors cited Liu et al. (2015, 2016) and Xiu et al. (2017b) to support their treatment of SOA-derived BrC, these studies actually demonstrated that biogenic SOA showed significantly lower light absorption compared to aromatic VOC-derived SOA. Indeed, this finding explains why previous studies, including Wang et al. (2014) and others cited in the introduction, specifically considered only aromatic SOA in their BrC calculations. Given the distinct spatial distribution patterns of biogenic and anthropogenic SOAs, the model simulations should be revised to treat only aromatic SOA as light-absorbing while explicitly treating biogenic SOA as non-absorbing.

**Response:** We have separated the aromatic SOA from the whole SOA produced from anthropogonic and biogenic sources in the model as suggested, setting the aromatic SOA as the secondary source of BrC. Additionally, since our study do not consider the dynamic variations of BrC absorption, we have added corresponding explanations in main the text. We have clarified in lines 176 to 185:

" SOA has also shown light absorption in the atmosphere (Lin et al., 2014). Laboratory experiments have revealed that most of the light-absorbing SOA is associated with aromatic SOA (Jacobson, 1999; Laskin et al., 2015; Li et al., 2020) and the absorption from biogenic SOA in the field has been found to be negligible (Washenfelder et al., 2015) . Therefore, here we assume aromatic derived SOA as secondary BrC in the model following previous studies (Jo et al., 2016; Wang et al., 2018).

Moreover, it is worth noting that both primary and SOA light absorption were shown to be dynamic, where BrC can be bleached when they undergo photodissociation (Forrister et al., 2015; Wong et al., 2019), or be darken by cloud and fog processing of aerosols (Moise et al., 2015; Lin et al., 2017; Cheng et al., 2020). These processes are not considered in this study yet. More detailed parameterization of the chemical aging of BrC are needed in future BrC models. "

**2 Comment:** L121-130: The methodology for deriving BrC emissions from primary sources lacks clear documentation. The treatment of RCC emissions is particularly unclear - the authors do not specify whether they assumed all RCC organic aerosols contribute to BrC. Similarly, their apparent assumption that all BB OA acts as BrC represents an oversimplification. Even in a study more than 10 years ago, Feng et al. (2013) used a non-unity fraction (92%) based on scientific justification (Chen and Bond, 2010). The current study requires similar scientific justification for these critical assumptions about BrC source contributions.

**Response:** We have re-established the proportions of BrC from various primary emission sources based on the "bottom-up" inventories. We have clarified in in lines 146 to 175 in the main text:

"    For the primary emission of BrC, previous BrC simulations have substituted it with a proportion of POA directly (Feng et al., 2013; Lin et al., 2014; Wang et al., 2014; Tuccella et al., 2020; Xu et al., 2024), derived it from the relationship between the burning efficiency and the observed aerosol light absorption (Jo et al., 2016; Zhu et al., 2021), or determined it through parameterization where BrC absorption is a function of the BC-to-OA emission ratio (Zhang et al., 2020). In the present work, we have calculated the primary BrC emissions based on the bottom-up OA emission inventory combined with reported annual BrC emissions from various primary sources, as shown in Table 1. Firstly, we collect the reported annual emissions of BrC from RCC, BB and FFs-TRA by using bottom-up inventory method. It should be noted that given the proximity of the study period (January 2014) to 2013, we use the emissions of BrC from RCC and BB in 2013 provided by Sun et al. (2017; 2021), which is 592 Gg and 712 Gg, respectively. The emissions of FFs-TRA derived BrC is 76Gg, which is calculated based on the value of 2017 (Wang et al., 2022) and scaled by a factor of 0.70 to reflect the change of annual civilian-owned motor vehicles. We assume that the spatial and seasonal variation of BrC is similar to OA. Then bottom-up emissions inventory induced monthly BrC emissions in the NCP in January 2014 is the annual BrC emissions multiplied by the ratio of OA emissions in the NCP *vs* China, and the ratio of OA emissions in January 2014 *vs* the whole year, resulting in a value of 65.5 Gg, 56.8 Gg and 4.4 Gg for RCC, BB and FFs-TRA, respectively. Finally, the proportion of the three primary emissions of BrC used in the model is 36.3%, 100.8% and 15.8%, respectively. Figure 2 shows the contributing regions and burdens of the three separated primary sources of BrC."

Table 1 The data for primary BrC emissions calculation

| Primary sources of BrC | RCC | BB | FFs-TRA |
|---|---|---|---|
| Annual BrC emissions (Gg) in China | 592.0[a] | 712.0[b] | 76.0[c] |
| Ratio of OA emissions in the NCP *vs* China[e] | 57.7% | 51.0% | 69.4% |
| Ratio of OA emissions in January 2014 *vs* the whole year[e] | 19.2% | 14.0% | 8.3% |
| Bottom-up emissions inventory induced monthly BrC emissions in the NCP in January | 65.5 | 56.8[d] | 4.4 |
| Emissions in the NCP in January 2014[e] | 180.2 | 56.4 | 27.9 |
| BrC emissions ratio for primary sources used in the model | 36.3% | 100.8% | 15.8% |

[a] The BrC emissions from China's RCC in 2013 was reported by Sun et al., (2017) based on experiments involving seven coals were burned in four typical stoves as both chunk and briquette styles.

[b] The calculated BrC emissions from China's household biomass burning in 2013 reported by Sun et al., (2021) using 11 widely used biomass types in China burned in a typical stove.

[c] The estimated BrC emissions from vehicle exhaust in 2017 was 109 Gg reported by Wang et al., (Wang et al., 2022). In this study, the emissions of FFs-TRA derived BrC is 76.0 Gg with a yearly scale factor 0.70 which derived by the annual civilian-owned motor vehicles between 2014 and 2017.

[d] The value of BrC emissions in NCP in January 2014 is additionally added with OA emitted from the open-biomass burning (6 Gg) which is assumed to be entirely light-absorbing.

[e] These values were derived from the OA emission inventory described in Sec. 2.1

[Figure]

Figure 2. Monthly BrC emissions burdens in January 2014 in the NCP from RCC, BB and FFs-TRA.

**3 Comment:** L148-154 & Table 1: The selection criteria for MAE and k of BrC require clarification. The authors inconsistently adopted MAE values from Zhang et al. (2022) for primary BrC while using values from Ni et al. (2021) for secondary BrC, although Zhang et al. (2022) reported both. Furthermore, the refractive indices from Ni et al. (2021) are notably lower than those reported in Zhang et al. (2022) (e.g., MAC at 370 nm: 2.39 for BBOA vs. 2.08 for LO-OOA) and other cited studies such as Xie et al. (2017) and Liu et al. (2016), where $k_{365}$ values for toluene SOA range from 0.008 (H2O2) to 0.025 (NOx). For consistency and

completeness, the study should adopt values from Zhang et al. (2022) for both primary and secondary BrC, which would also ensure consistent wavelength ranges (370-660 nm) across all BrC components.

**Response:** We have consistently adopted MAE values from results of Zhang (2022b) as the HI-BRC-ABS case. Additionally, a series low MAE have been selected as the LOW-BRC-ABS. We have clarified in Lines 197-213 in the main text:

"    In this study, as shown in Table 2, two sets of MAE are used for the sensitivity experiments of BrC. We choose a field optical measurement of BrC from all sources made by Zhang et al. (2022b) as the high absorption case (HI-BRC-ABS). The optical properties of BB and FFs-TRA obtained in laboratory by Xie et al. (2017), as well as MAE of RCC and secondary BrC obtained in laboratory by Ni et al. (2021) are adopted as the low absorption case (LOW-BRC-ABS) in the study. The imaginary part of the two cases have shown wavelength dependent light-absorption properties and the changes in anthropogenic emissions affect the optical properties of BrC. The imaginary part of both two cases are interpolated to 11 wavelengths to match the aerosol radiation calculation of Goddard module in the WRF-Chem model. The value of $k$ in this work is derived from the measured MAE using the following Eq.(6) (Liu et al., 2013; Lu et al., 2015) as shown in Table 2:

$$k_{BrC,\lambda} = \frac{\rho \times \lambda \times MAE_\lambda}{4\pi} \tag{6}$$

Where $MAE_\lambda$ (m$^2$ g$^{-1}$) is the bulk mass absorption efficiency of BrC at the corresponding wavelength $\lambda$. $\rho$ (g cm$^{-3}$) is the density of organic aerosols, which is assigned as 1.2 g cm$^{-3}$ (Turpin and Lim, 2001) in this study."

Table 2 The refractive index of BrC used in the model

| Aerosols | Wavelength (nm) | k values for HI-BRC-ABS | k values for LOW-BRC-ABS |
|---|---|---|---|
| BrC-RCC | 365 | - | 0.0320 |
| | 370 | 0.1890 | - |
| | 470 | 0.0608 | - |
| | 500 | - | 0.0020 |
| | 520 | 0.0272 | - |
| | 590 | 0.0173 | - |
| | 660 | 0.0081 | - |
| BrC-BB | 365 | - | 0.0300 |
| | 370 | 0.0587 | - |
| | 405 | - | 0.0016 |
| | 470 | 0.0219 | - |
| | 520 | 0.0120 | - |
| | 550 | - | 0.0026 |
| | 590 | 0.0092 | - |
| | 660 | 0.0046 | - |
| BrC-FFs-Tra | 365 | - | 0.0180 |
| | 370 | 0.0509 | - |
| | 405 | - | 0.0130 |
| | 470 | 0.0194 | - |
| | 520 | 0.0085 | - |
| | 550 | - | 0.0045 |
| | 590 | 0.0046 | - |
| | 660 | 0.0018 | - |
| BrC-SOA | 365 | - | 0.0049 |
| | 370 | 0.0251 | - |
| | 470 | 0.0166 | - |
| | 500 | - | 0.0007 |
| | 520 | 0.0114 | - |
| | 590 | 0.0107 | - |
| | 660 | 0.0063 | - |

'-' means not available

**4 Comment:** L55-57: The claim of "a growing number of studies" is not supported by the cited references. One reference is from 2001, and the 2020 reference actually states that "fossil-fuel combustion is not an important BrC emitter". Given that fossil-fuel BrC is an important component of this study, the introduction requires a more comprehensive discussion of how the scientific understanding of fossil-fuel BrC has evolved. This should include an explanation of why fossil-fuel BrC was historically overlooked, and how recent observational and modeling studies have revealed its importance. While some of these aspects are mentioned in L57-67,

the current discussion lacks clear organization and logical flow, and should be restructured to present this evolution more coherently.

**Response:** As suggested, we have reorganized the document to support the claim that why fossil-fuel BrC, especially RCC derived BrC should be taken seriously in China in lines 44-70 in the main text:

" It has been well established that BrC is not a single substance, but a general term for light-absorbing organic aerosols. A series of laboratory measurements and observations in the earlier years demonstrate that BrC is mainly associated with smoldering biomass burning (BB) or biofuel (BFs) combustion (Chakrabarty et al., 2010; Chen and Bond, 2010; Lack et al., 2012; Washenfelder et al., 2015; Kumar et al., 2018). On the other hand, OA from fossil-fuel combustion are generally assumed to be non-absorbing as the combustion conditions for fossil fuels (FFs) are typically not conducive for BrC formation (Hecobian et al., 2010; Shapiro et al., 2009; Bond et al., 2013). Therefore, earlier climate model studies have assumed that primary OA from BB and BFs combustion is the main or sole BrC source (Feng et al., 2013; Jacobson, 2014; Saleh et al., 2014; Hammer et al., 2016; Brown et al., 2018). Recent studies have also incorporated the ageing of secondary organic aerosol (SOA) (Jo et al., 2016; Wang et al., 2018; Zhang et al., 2020). However, more recent exceptions are being found in low-efficiency residential-coal combustion (RCC) (Bond, 2001; Yan et al., 2017; Xie et al., 2019; Tian et al., 2019; Zhang et al., 2022a) and fuel-oil combustion in vehicle and ship engines (Xie et al., 2017; Corbin et al., 2019; Tang et al., 2020; Huang et al., 2022). It is now generally accepted that the formation of BrC is not exclusively linked to the chemical make-up of biomass fuels but is most critically determined by the combustion conditions (Saleh et al., 2018; Cheng et al., 2020; Saleh, 2020; Wang et al., 2022). The key factor contributing to the high levels of BrC observed from biomass fuels is their combustion under relatively low-temperature and fuel-rich conditions, which are highly favorable for BrC formation. In contrast, fossil fuels, such as those burned in internal combustion engines, typically undergo combustion at higher temperatures and under more fuel-lean conditions, which are less conducive to BrC production (Saleh, 2020). China, as a developing country, coal is commonly used for residential heating in cold season, causing massive emissions of organic particles (Yan et al., 2017; Li et al., 2018). According to the National Bureau of Statistics of China (https://data.stats.gov.cn), the coal consumption in 2014 was about 4000 Tg, accounting for 65.8% of the total primary energy use of China. Of this, around 93 Tg is used as household fuel. The poor burning conditions and limited emission control facilities in this region could lead to substantial

emissions of BrC. This could explain why, to date, all reported instances of coal-derived BrC have originated from China. Both Yan et a., (2017) and Mo et al., (2021) have used dual carbon isotope-based source apportionment method reported that fossil fuel, especially coal combustion from the residential sector is important source in northern China, even the largest contributor in some regions."

**5 Comment:** L54-55: This limitation (BrC from FFs) usually extends to chemical transport models and atmospheric chemistry models as well. Climate models present an even greater concern, as they typically do not consider brown carbon at all. And these statements require supporting references from the literature.

**Response:** As suggested, we have added references for supporting the statements. We have clarified in Lines 71-73 in the main text:

"These recent findings indicate a critical gap on the treatment of BrC in chemical transport models, atmospheric chemistry models and climate models as well, which present an even greater concern, as they typically do not consider BrC at all (Ma et al., 2021; Jo et al., 2023; Gao et al., 2025; Ge et al., 2025)."

**6 Comment:** L101: A reference for the density values (1.2 g cm-3 for primary and 1.0 g cm-3 for secondary BrC) is needed.

**Response:** We have included the references for the density values in lines 134 to 135:
"The BrC in the model has an effective density of 1.2 g cm$^{-3}$ for primary BrC (Turpin and Lim, 2001) and of 1.0 g cm$^{-3}$ for secondary BrC (Hurley et al., 2001)."

**7 Comment:** L113: I would recommend rephrasing this sentence, such as "To date, studies on BrC emissions have been limited"

**Response:** We have rephrased sentence as suggested.

**8 Comment:** L124: The manuscript contains numerous grammatical errors that need attention. Professional language editing is recommended. Some examples include:
• L125: "which resulting" should be "resulting in~"
• L145: "it follow the study" should be "this study follows~"

**Response:** We have made corresponding corrections as suggested and conducted systematically and comprehensive revisions marked in yellow in the revised manuscript.

**9 Comment:** L191: The reference "Bai et al. (2022)" is missing from the reference list. Additionally, the LGHAP dataset requires further explanation regarding its derivation methodology and validation.

**Response:** We have included the reference in the reference list and also double-checked the reference list. We have also included the explanation regarding derivation methodology and validation of LGHAP dataset in Lines 266-270 in the main text:

"This gap-free daily AOD dataset at 1 km resolution for 2000–2020 in China is generated by integrating multimodal data from satellites, numerical models, and in situ measurements. Data gaps in Moderate Resolution Imaging Spectroradiometer (MODIS) AOD are reconstructed through spatial pattern recognition and statistical knowledge transfer. Validation against Aerosol Robotic Network (AERONET) observations shows strong agreement, with an R of 0.91 and an RMSE of 0.21."

**10 Comment:** L204: The authors need to provide details on how the model calculates hygroscopic growth for different aerosol components. This is particularly important as hygroscopic growth significantly affects the model's AOD calculations and subsequent comparisons with satellite observations.

**Response:** We have clarified in lines 124-133 in the main text:

"It is worth noting that the aerosol liquid water content in the study is predicted with the inorganic aerosols using a computationally efficient thermodynamic equilibrium model, ISORROPIA (version 1.7, Nenes et al., 1998; Fountoukis and Nenes, 2007). In this study, ISORROPIA is mainly used to predict the thermodynamic equilibrium between the ammonium-sulfate-nitrate-chloride-water aerosols and their gas-phase precursors $H_2SO_4$-$HNO_3$-$NH_3$-HCL-water vapor, and water uptake of aerosols is calculated using the Zdanovskii-Stokes-Robinson (ZSR) correlation (Stokes and Robinson, 1966):

$$W = \sum_i \frac{M_i}{m_{oi}(a_w)} \tag{5}$$

Where $W$ is the mass concentration of aerosol liquid water (kg m$^{-3}$ air), $M_i$ is the molar concentration of species $i$ (mol m$^{-3}$ air), and $m_{oi}(a_w)$ is the molality of an aqueous binary solution of the $i$-th electrolyte with the same $a_w$ (i.e. relative humidity) as in the multicomponent solution."

**11 Comment:** L215-223: SSA measurements typically have larger uncertainties at low AOD values. Did the authors apply any screening criteria for SSA values associated with low AOD conditions? If not, the authors may want to do a screening to understand the SSA simulation uncertainty.

**Response:** As suggested, we have applied a screening criterion to exclude data points with AOD < 0.05 (at 440 nm) to mitigate the influence of high measurement uncertainties. The post-screening results shown improvement between modeled and observed SSA values with enhanced correlation coefficients (R from 0.49 to 0.54). The revised figure and updated results are detailed in lines 249-263 (Please refer to **Response** to **Comment 13**).

**12 Comment:** L241-252: The reported values in this section are meaningless without proper scientific justification for the key assumptions as mentioned in earlier comments:
• The BrC ratios from different primary sources
• The arbitrary 10% SOA contribution to BrC

**Response:** We have addressed the issues as described in the **Response** to **Comment 1** and **Comment 2**. Additionally, based on the revisions to emission sources, we have correspondingly updated the calculation results in Lines 310-319 in the main text. Due to changes of emission sources, the overall concentration of total BrC shows a slight increase. This is manifested by an increased proportion of RCC derived BrC and a reduction in FFs-TRA sources, while the proportions of BB and secondary BrC remain generally unchanged.

**13 Comment:** L254-273: The SOA-BrC values reported here should be higher if k values from other studies were adopted, as discussed in the previous comments. While sensitivity simulations were mentioned in the introduction, no such analyses appear in the manuscript.

This section would benefit from sensitivity tests exploring the impact of different k values on the results.

**Response:** We really appreciate the reviewer's suggestion and have conducted three sensitivity experiments using different *k* values. We have clarified in Lines 241-263 in the main text: "SSA determines the strength of aerosols in absorbing solar radiation. Here we conduct three sensitivity experiments to evaluate the effect of BrC with different *k* values on the simulated aerosol absorption. The first experiment is the control simulation in which all organic aerosols are treated as purely scattering particles with no absorption contribution of BrC, which is referred to as NOBRC. The hi-absorption scenario (HI-BRC-ABS) and low-absorption scenario (LOW-BRC-ABS) characterize BrC light absorption by using the higher and lower imaginary refractive index derived from Section 2.3.2, respectively. Figure 3 shows the comparisons of simulated versus observed SSA at 440 nm (SSA$_{440}$) at Sun-sky radiometer Observation NETwork (SONET) sites in Beijing, Songshan, Xi'an, Hefei, and Nanjing in January 2014. Due to the influence of clouds, the observational data from SONET are not continuous, resulting in a total of 237 valid data points are available for comparisons. Moreover, SSA retrieval typically have larger uncertainties at low AOD values (Dubovik et al., 2002). Therefore, we have excluded the SSA data when AOD is less than 0.5, which has 206 valid points in each case. We find that the inclusion of BrC in the model reduces the bias of simulated SSA. The HI-BRC-ABS case demonstrated a largest improvement with the correlation coefficient increasing to 0.54, making it the best simulation in the study. It suggests that stronger BrC absorption case, as prescribed in HI-BRC-ABS, better captures the aerosol optical properties observed in northern China during the winter. Consequently, the HI-BRC-ABS case can serve as the base simulation for further investigation of radiative effects of BrC in this study. Overall, the model tends to underestimate SSA$_{440}$. The underestimation might be partly caused by the overestimation of absorbing aerosols like BC or dust. Meanwhile, the uncertainties of the simulated SSA can be caused by other factors, such as mixing state of aerosols, particle shape, wavelength, and mass ration of non-black carbon to BC (Liu et al., 2017; Jeong et al., 2020)."

[Figure]

Figure 3. Scatter plot and linear fitting of modelled and observed column integrated SSA at 440 nm in the HI-BRC-ABS (red), LOW-BRC-ABS (blue) and NOBRC (black) case.

**14 Comment:** L269-271: The attribution of differences to aerosol density and mixing state lacks sufficient discussion and scientific justification. A more likely explanation involves the vertical distribution of aerosols: while the authors compare surface concentrations of primary versus secondary BrC, AOD represents column-integrated concentrations. Given that SOA/POA ratios typically increase with altitude, analysis of vertical profiles should help clarify these differences.

**Response:** We have provided the vertical profile of the domain average ratio of SOA to POA during the simulation period and clarified in Lines 336-345 in the main text:

"Despite lower surface concentrations compared to primary BrC, secondary BrC contributes significantly to AAOD of BrC, averaging ~10.0% with elevated contributions in the sea and remote regions, which is likely due to the highly oxidized character of organic aerosols and its chemical aging in aging air masses leading to the formation of BrC (Gouw et al., 2005; Kawamura et al., 2005; Tsigaridis and Kanakidou, 2018). While AOD represents column-integrated concentrations, the secondary BrC to Primary BrC ratio increases from 8.9% at the surface to 12.0% of atmospheric burden. It reaches 14.3% at an altitude of 500m as shown in Figure S6, which could lead to a higher absorption contribution of secondary BrC (Wang et al., 2022b). Moreover, the observations indicate that a substantial SOA is water-soluble (Maria et

al., 2003; Peng et al., 2021) which is treated as hygroscopic components in the model and its absorption could be magnified."

**15 Comment:** L310-313: This pattern likely stems from the authors' arbitrary assumption that 10% of total SOA contributes to BrC. This assumption incorrectly attributes a significant portion of biogenic SOA from southern China as BrC, which does not reflect actual atmospheric conditions.

**Response:** We have redefined the secondary BrC as suggested (Please refer to the **Response** to **Comment 1**). The persistent high values over southern China might stem from the model's representation of secondary BrC as hygroscopic components, whose light-absorbing capacity is amplified in the region with high ambient humidity. We have clarified in Lines 385-387: "The persistent high values over southern China might stem from the model's representation of secondary BrC as hygroscopic components (Peng et al., 2021), whose light-absorbing capacity is amplified in the region with high ambient humidity."

**16 Comment:** References: Several references are incomplete. For example, the journal name and DOI are not available in Feng et al. (2013). The authors should double-check the reference list.

**Response:** We have double-checked the reference list.

**In addition to** the comments raised by the reviewer above, we have also reorganized the abstract of the manuscript and made some additional explanations in Lines 413-418 in the main text:

[revised manuscript text omitted]

---

## Author Comment (AC2)

**Reply to Anonymous Referee #2**

We thank the reviewer for the careful reading of the manuscript and helpful comments. We have revised the manuscript following the suggestion, as described below.

**Major comments:**

**1 Comment:** While the authors provided many details in the methodology, there are still some missing important information. For example, BrC refractive index and DRE calculations are well explained, but AOD calculations are not mentioned in the methodology. Since the modeled AOD is discussed and compared with observations, the author should briefly describe the process of calculating AOD (e.g., Mie theory uses refractive indices as inputs?)

**Response:** As suggested, we have included Section "2.2 Aerosol radiative module" in Lines 98-123 in the main text:

"2.2 Aerosol radiative module

The aerosol radiative module developed by Li et al. (2011) has been incorporated into the WRF-Chem model to calculate the aerosol optical depth (AOD or $\tau_a$), single scattering albedo (SSA or $\omega_a$), and the asymmetry factor ($g_a$). In the aerosol module, aerosols are represented by a three-moment approach with a lognormal size distribution:

$$n(lnD) = \frac{N}{\sqrt{2\pi}ln\sigma_g} exp[-\frac{1}{2}(\frac{lnD-lnD_g}{ln\sigma_g})^2] \tag{1}$$

Where D is the particle diameter, N is the number distribution of all particles in the distribution, $D_g$ is the geometric mean diameter, and $\sigma_g$ is the geometric standard deviation. To calculate the aerosol optical properties, the aerosol spectrum is divided into 48 bins from 0.002 to 20.0 μm, with radius $r_i$. The aerosols are classified into four types: (1) internally mixed sulfate, nitrate, ammonium, hydrophilic organics and black carbon (BC), and water; (2) hydrophobic organics; (3) hydrophobic BC; and (4) other unidentified aerosols (generally dust-like aerosols). These four kinds of aerosols are assumed to be mixed externally. For the internally mixed aerosols, the complex refractive index at a certain wavelength ($\lambda$) is calculated based on the volume-weighted average of the individual refractive index. Given the particle size and complex refractive index, the extinction efficiency ($Q_e$), $\omega_a$ and $g_a$ are calculated using the Mie theory at a certain wavelength ($\lambda$). The look-up tables of $Q_e$, $\omega_a$ and $g_a$ are established according to particle sizes and refractive indices to avoid multiple Mie scattering calculation.

The aerosol optical parameters are interpolated linearly from the look-up tables with the calculated refractive index and particle size in the module. The $\tau_a$ at a certain $\lambda$ in a given atmospheric layer $k$ is determined by the summation over all types of aerosols and all bins:

$$\tau_a(\lambda, k) = \sum_{i=1}^{48} \sum_{j=1}^{4} Q_e(\lambda, r_i, j, k) \pi r_i^2 n(r_i, j, k) \Delta Z_k \tag{2}$$

where $n(r_i, j, k)$ is the number concentration of $j$-th kind of aerosols in the $i$-th bin. $\Delta Z_k$ is the depth of an atmospheric layer. The weighted-mean values of $\omega_a$ and $g_a$ are then calculated by using D'Almeida et al., (1991):

$$\omega_a(\lambda, k) = \frac{\sum_{i=1}^{48} \sum_{j=1}^{4} Q_e(\lambda, r_i, j, k) \pi r_i^2 n(r_i, j, k) \omega_a(r_i, j, k) \Delta Z_k}{\sum_{i=1}^{48} \sum_{j=1}^{4} Q_e(\lambda, r_i, j, k) \pi r_i^2 n(r_i, j, k) \Delta Z_k} \tag{3}$$

$$g_a(\lambda, k) = \frac{\sum_{i=1}^{48} \sum_{j=1}^{4} Q_e(\lambda, r_i, j, k) \pi r_i^2 n(r_i, j, k) \omega_a(r_i, j, k) g_a(\lambda, r_i, j, k) \Delta Z_k}{\sum_{i=1}^{48} \sum_{j=1}^{4} Q_e(\lambda, r_i, j, k) \pi r_i^2 n(r_i, j, k) \omega_a(r_i, j, k) \Delta Z_k} \tag{4}$$

When the wavelength-dependent $\tau_a$, $\omega_a$, and $g_a$ are calculated, they can be used in the Goddard shortwave module."

**2 Comment:** Any assumption made in the study needs either a reference or an explanation. There are multiple assumptions in your study that are not justified. For example, "In this work, a proportion of 10% of the total SOA is included as a part of BrC" or "The BrC in the model has an effective density of 1.2 g cm-3 for primary BrC and of 1.0 g cm-3 for secondary BrC".

**Response:** We have separated the aromatic SOA from the whole SOA produced from anthropogonic and biogenic sources in the model as suggested, setting the aromatic SOA as the secondary source of BrC. Additionally, since our study do not consider the dynamic variations of BrC absorption, we have added corresponding explanations. We have clarified in Lines 176-185 in main the text:

"   SOA has also shown light absorption in the atmosphere (Lin et al., 2014). Laboratory experiments have revealed that most of the light-absorbing SOA is associated with aromatic SOA (Jacobson, 1999; Laskin et al., 2015; Li et al., 2020) and the absorption from biogenic SOA in the field has been found to be negligible (Washenfelder et al., 2015) . Therefore, here we assume aromatic derived SOA as secondary BrC in the model following previous studies (Jo et al., 2016; Wang et al., 2018).

    Moreover, it is worth noting that both primary and SOA light absorption were shown to be dynamic, where BrC can be bleached when they undergo photodissociation (Forrister et al., 2015; Wong et al., 2019), or be darken by cloud and fog processing of aerosols (Moise et al., 2015; Lin et al., 2017; Cheng et al., 2020). These processes are not considered in this study yet.

More detailed parameterization of the chemical aging of BrC are needed in future BrC models. "

We have added the references for the BrC effective density and clarified in Lines 134-135:
"The BrC in the model has an effective density of 1.2 g cm$^{-3}$ for primary BrC (Turpin and Lim, 2001) and of 1.0 g cm$^{-3}$ for secondary BrC (Hurley et al., 2001)."

**Minor comments:**

**1 Comment:** Page 6, line 125: "RCC is responsible for about 45% of primary BrC emissions". Is this percentage based on your calculations? Or from other studies or data? Please use proper citations or discuss the calculations further.

**Response:** We have re-established the proportions of BrC from various primary emission sources based on the "bottom-up" inventories. We have clarified in Lines 146-175 in the main text:

"    For the primary emission of BrC, previous BrC simulations have substituted it with a proportion of POA directly (Feng et al., 2013; Lin et al., 2014; Wang et al., 2014; Tuccella et al., 2020; Xu et al., 2024), derived it from the relationship between the burning efficiency and the observed aerosol light absorption (Jo et al., 2016; Zhu et al., 2021), or determined it through parameterization where BrC absorption is a function of the BC-to-OA emission ratio (Zhang et al., 2020). In the present work, we have calculated the primary BrC emissions based on the bottom-up OA emission inventory combined with reported annual BrC emissions from various primary sources, as shown in Table 1. Firstly, we collect the reported annual emissions of BrC from RCC, BB and FFs-TRA by using bottom-up inventory method. It should be noted that given the proximity of the study period (January 2014) to 2013, we use the emissions of BrC from RCC and BB in 2013 provided by Sun et al. (2017; 2021), which is 592 Gg and 712 Gg, respectively. The emissions of FFs-TRA derived BrC is 76Gg, which is calculated based on the value of 2017 (Wang et al., 2022) and scaled by a factor of 0.70 to reflect the change of annual civilian-owned motor vehicles. We assume that the spatial and seasonal variation of BrC is similar to OA. Then bottom-up emissions inventory induced monthly BrC emissions in the NCP in January 2014 is the annual BrC emissions multiplied by the ratio of OA emissions in the NCP *vs* China, and the ratio of OA emissions in January 2014 *vs* the whole year, resulting in a value of 65.5 Gg, 56.8 Gg and 4.4 Gg for RCC, BB and FFs-TRA, respectively. Finally, the proportion of the three primary emissions of BrC used in the model is 36.3%, 100.8% and

15.8%, respectively. Figure 2 shows the contributing regions and burdens of the three separated primary sources of BrC."

Table 1 The data for primary BrC emissions calculation

| Primary sources of BrC | RCC | BB | FFs-TRA |
|---|---|---|---|
| Annual BrC emissions (Gg) in China | 592.0[a] | 712.0[b] | 76.0[c] |
| Ratio of OA emissions in the NCP *vs* China[e] | 57.7% | 51.0% | 69.4% |
| Ratio of OA emissions in January 2014 *vs* the whole year[e] | 19.2% | 14.0% | 8.3% |
| Bottom-up emissions inventory induced monthly BrC emissions in the NCP in January | 65.5 | 56.8[d] | 4.4 |
| Emissions in the NCP in January 2014[e] | 180.2 | 56.4 | 27.9 |
| BrC emissions ratio for primary sources used in the model | 36.3% | 100.8% | 15.8% |

[a] The BrC emissions from China's RCC in 2013 was reported by Sun et al., (2017) based on experiments involving seven coals were burned in four typical stoves as both chunk and briquette styles.

[b] The calculated BrC emissions from China's household biomass burning in 2013 reported by Sun et al., (2021) using 11 widely used biomass types in China burned in a typical stove.

[c] The estimated BrC emissions from vehicle exhaust in 2017 was 109 Gg reported by Wang et al., (Wang et al., 2022). In this study, the emissions of FFs-TRA derived BrC is 76.0 Gg with a yearly scale factor 0.70 which derived by the annual civilian-owned motor vehicles between 2014 and 2017.

[d] The value of BrC emissions in NCP in January 2014 is additionally added with OA emitted from the open-biomass burning (6 Gg) which is assumed to be entirely light-absorbing.

[e] These values were derived from the OA emission inventory described in Sec. 2.1

[Figure]

Figure 2 Monthly BrC emissions burdens in January 2014 in NCP from RCC, BB and FFs-TRA.

**2 Comment:** Page 7, line 148: Expand MAE when it appears for the first time. This applies to all other abbreviations used in the manuscript.

**Response:** We have explained MAE in Lines 195-196 and double checked other abbreviations in the main text and Supplement.

**3 Comment:** Page 9: Give more details on LGHAP and OMI satellites (e.g., horizontal resolution, local time overpass, etc.). Then discuss those further. For example, are you comparing model results at the satellite overpass time?

**Response:** We have added detailed descriptions of both the LGHAP in Lines 260-264 and OMI in Lines 272-274, respectively:

Lines 266-270: "This gap-free daily AOD dataset at 1 km resolution for 2000–2020 in China is generated by integrating multimodal data from satellites, numerical models, and in situ measurements. Data gaps in Moderate Resolution Imaging Spectroradiometer (MODIS) AOD are reconstructed through spatial pattern recognition and statistical knowledge transfer. Validation against Aerosol Robotic Network (AERONET) observations showed strong agreement, with an R of 0.91 and an RMSE of 0.21."

Lines 278-280: "OMI aboard NASA's Aura satellite offers global atmospheric measurements at a spatial resolution of 0.25°×0.25°, with Beijing's overpass occurring at approximately 13:45 local time. The average AOD440 from the model simulation at 14:00 local time shows generally agreement with the OMI retrieval."

**4 Comment:** Page 10, line 204: Change "hydroscopic" to "hygroscopic". Hygroscopic growth is the size increase of aerosols due to water vapor absorption.

**Response:** We have changed "hydroscopic" to "hygroscopic" in Line 284 in the main text.

**5 Comment:** Page 11, line 221: Change "ration" to "ratio".

**Response:** We have rephrased the sentence as "The underestimation might be partly caused by the overestimation of absorbing aerosols like BC or dust" in Lines 257-258 in the main text.

**6 Comment:** Page 13, line 259: Citation is needed for "Compared with the global mean ratio of BrC to BC which is 1.24".

**Response:** We have added the citation in Lines 326-328 in the main text.

**7 Comment:** Page 15, line 288: "the increased $DRE_{TOA}$ induced by BrC which is usually considered as its scattering effect is up to an average of +0.37 W m-2 …". This sentence is confusing a little. The authors should clarify if they are talking about BrC's net radiative effect or just absorption. As mentioned in Page 15, line 285: "the BrC populations have a net cooling effect". So, the net BrC DRE could not increase OA DRE.

**Response:** We agree with the comment and have clarified in Line 404 in the main text: "the absorption of BrC increases the $DRE_{TOA}$ of OA by 28.0% with an average of +0.40 W m$^{-2}$ and a maximum of +1.83 W m$^{-2}$"

**8 Comment:** Figures 8 and 9: Panel letter C is missing.

**Response:** We have updated Figures 8 and 9.

**9 Comment:** Figure S1: Adding error bars to the time series would be helpful since the authors are demonstrating average concentrations over all the monitors in NCP.

**Response:** We have updated Figure S1 as suggested.

**10 Comment:** Figure S4: The units inside the charts (MB and RMSE) should be updated.

**Response:** We have updated the units (MB and RMSE) inside the charts as suggested.

In addition to the questions raised by the reviewer above, we have also reorganized the abstract of the manuscript and made some additional explanations in Lines 413-418 in the main text: "It should be noted that China has started to switch from coal to cleaner and more efficient energy such as natural gas or liquid petroleum gas in recent years. According to the latest report of National Bureau of Statistics of China, the total coal consumption for residential use is 55.5 Gg in 2022 (https://data.stats.gov.cn) with a 40.3% decrease compared to 2014. Therefore, our diagnosis of the sources of BrC and their radiative effects is specifically targeted at the winter season in 2014. Moreover, future simulations should strengthen the parameterization for the evolution of BrC, such as bleaching or darkening processes."

Furthermore, we have conducted additional sensitivity experiments on different *k* values.

NOBRC: All organic aerosols treated as purely scattering (no BrC absorption);

LOW-BRC-ABS: BrC absorption with lower *k* values;

HI-BRC-ABS: BrC absorption with higher *k* values.

Comparisons of modeled and observed SSA across these experiments demonstrate that the HI-BRC-ABS configuration yields the closest agreement with observations. We have clarified in Lines 241-263 in the main test:

[revised manuscript text omitted]